# Recovery of the full in vivo firing range in post-lesion surviving DA SN neurons associated with Kv4.3-mediated pacemaker plasticity

Lora Kovacheva[1]*, Josef Shin[1,2], Josefa Zaldivar-Diez[1,3], Johanna Mankel[1], Navid Farassat[1,4], Kaue Machado Costa[1,5], Poonam Thakur[1,6], José A Obeso[3], Jochen Roeper[1]*

[1]Institute for Neurophysiology, Neuroscience Center, Goethe University Frankfurt, Frankfurt, Germany; [2]Department of Neurology, University Medical Centre of the Johannes Gutenberg University Mainz, Mainz, Germany; [3]Neuroscience Center at Fundación de Investigación HM Hospitales, CEU San Pablo University, Madrid, Spain; [4]Eye Center, Medical Center-University of Freiburg, Freiburg im Breisgau, Germany; [5]Department of Psychology, University of Alabama, Birmingham, United States; [6]School of Biology, Indian Institute of Science Education and Research (IISER), Thiruvananthapuram, India

**\*For correspondence:**
lorask@gmail.com (LK);
roeper@em.uni-frankfurt.de (JR)

**Competing interest:** The authors declare that no competing interests exist.

**Abstract** Dopamine (DA) neurons in the substantia nigra (SN) control several essential functions, including the voluntary movement, learning, and motivated behavior. Healthy DA SN neurons show diverse firing patterns in vivo, ranging from slow pacemaker-like activity (1–10 Hz) to transient high-frequency bursts (<100 Hz), interspersed with pauses that can last hundreds of milliseconds. Recent in vivo patch experiments have started to reveal the subthreshold mechanisms underlying this physiological diversity, but the impact of challenges like cell loss on the in vivo activity of adult DA SN neurons, and how these may relate to behavioral disturbances, is still largely unknown. We investigated the in vivo electrophysiological properties of surviving SN DA neurons after partial unilateral 6-OHDA lesions, a single-hit, non-progressive model of neuronal cell loss. We show that mice subjected to this model have an initial motor impairment, measured by asymmetrical rotations in the open field test, which recovered over time. At 3 weeks post-lesion, when open field locomotion was strongly impaired, surviving DA SN neurons showed a compressed in vivo dynamic firing range, characterized by a 10-fold reduction of in vivo burst firing compared to controls. This in vivo phenotype was accompanied by pronounced in vitro pacemaker instability. In contrast, in the chronic post-lesion phase (>2 months), where turning symmetry in open field locomotion had recovered, surviving SN DA neurons displayed the full dynamic range of in vivo firing, including in vivo bursting, similar to controls. The normalized in vivo firing pattern was associated with a twofold acceleration of stable in vitro pacemaking, mediated by Kv4.3 potassium channel downregulation. Our findings demonstrate the existence of a homeostatic pacemaker plasticity mechanism in surviving DA SN neurons after pronounced cell loss.

## Editor's evaluation

Kovacheva et al. investigated the electrophysiological adaptations of surviving substantia nigra dopaminergic neurons at 3 weeks and over 2 months after a 6-OHDA lesion in a Parkinsonism mouse model. They report an important early loss of burst firing and pacemaker stability, followed

by a recovery of firing activity and a twofold increase in pacemaking at later stages. These compelling evidences are linked to the downregulation of the Kv4.3 potassium channel. The work provides significant insights into neuronal plasticity and will be of interest to basic neuroscientists and Parkinson's disease researchers.

## Introduction

Midbrain dopamine (DA) neurons in the substantia nigra (SN) project mainly to the dorsal striatum (DS), where they release DA in an activity-dependent manner, further shaped by additional mechanisms at the distal axon (*Liu et al., 2022*, *Kramer et al., 2022*). The resulting temporal and spatial dynamics of DA release in DS are relevant for multiple functions, including the control of voluntary movements, as evidenced by the cardinal features of Parkinson disease (PD), which is characterized by the death of SN DA neurons. The in vivo activity pattern of healthy SN DA neurons is complex, ranging from slow pacemaker-like activity (1–10 Hz) to transient high-frequency bursts (<100 Hz), and interspersed with pauses that can last hundreds of milliseconds. However, our understanding of and how SN DA neuron electrical activity adapts to cell loss, relevant during aging or PD, is limited.

We have previously shown that over expression of mutant A53T-α-synuclein induced an age-dependent acceleration of in vivo and in vitro firing frequencies. This was associated with redox-mediated dysfunction of Kv4.3 channels and an upregulation of Kv4.3 expression, potentially representing a homeostatic attempt to regulate pacemaker frequency (*Subramaniam et al., 2014a*). This study, however, left a more fundamental question unanswered: do adult DA SN neurons have the ability for homeostatic regulation of firing pattern in response to challenge? For this purpose, we employed the well-established 6-hydroxy-dopamine (6-OHDA) chemical lesion model in order to induce a unilateral loss of about half of the DA SN neurons. 6-OHDA has been widely used for generating non-progressive loss of DA SN neurons (*Rubi and Fritschy, 2020*; *Parker et al., 2016*; *Ungerstedt, 1968*). We opted for a unilateral, partial, intrastriatal 6-OHDA model, previously well characterized by *Bez et al., 2016*. Bez and colleagues, as well as others, demonstrated that these partial 6-OHDA DA lesions are non-progressive and follow a stereotypical time course: an initial damage phase dominated by cell loss, inflammation, and behavioral impairment for about 3 weeks, which is followed by a chronic phase (studied for up to 20 months in mice), characterized by partial behavioral recovery, neurochemical and molecular adaptations, and – to a certain degree – axonal sprouting of remaining DA neurons (*Bez et al., 2016*; *Cenci and Björklund, 2020*; *Kirik et al., 1998*; *Schwarting and Huston, 1996*; *Winkler et al., 2002*). This framework, with its distinct phases of impairment and subsequent partial functional recovery, provides a well-suited platform to investigate if and how the in vivo firing properties of surviving DA SN neurons change over time. Previous electrophysiological in vivo investigations of viable post-6-OHDA DA neurons have been limited, with only Hollerman and Grace providing a pioneering dataset (*Hollerman and Grace, 1990*). They found lesion-size-dependent changes in in vivo firing properties of putative DA neurons in rats, including a decrease of in vivo burst firing occurring 1 week after lesion, with larger lesions exacerbating this effect (*Hollerman and Grace, 1990*). Further supporting this, *Berretta et al., 2005* proposed that early electrophysiological changes in surviving DA neurons may result directly from 6-OHDA toxicity triggering a cascade of cellular events, such as hyperpolarization mediated by K-ATP channel activation. These electrophysiological adaptations occur within a broader neurodegenerative and neuroinflammatory timeline in the first 2 weeks post-lesion (*Stott and Barker, 2014*; *Walsh et al., 2011*). Importantly, in this study, investigations are conducted at two distinct time points – both after the stabilization of the early neuroinflammatory response – yet within a phase where behavioral dynamics are still evolving. This approach allows for a more refined analysis of homeostatic plasticity in identified DA SN neurons, capturing adaptations that persist beyond acute neurotoxicity while remaining within a window of functional and compensatory changes.

Here, we explore the electrophysiological properties, both in vivo and in vitro, of surviving SN DA neurons in an intra-striatal unilateral 6-OHDA lesion mouse model, titrated to about 40% surviving SN DA neurons. We selected two time points for recordings based on the continuous behavioral characterization of the motor phenotypes.

## Results

### Early behavioral impairments and long-term adaptations after partial lesion of substantia nigra dopamine neurons

To characterize electrophysiological properties in surviving and identified SN DA neurons, we employed a C57Bl6N mouse model with a partial 6-OHDA lesion of the nigrostriatal pathway. We explored different 6-OHDA concentrations (0.2–2 µg/µl) and volumes (2 µl and 6 µl), resulting in a range of 0.4–12 µg 6-OHDA to titrate the ipsilateral loss of DA SN neurons (*Figure 1—figure supplement 1* for details). To target 40–50% cell loss, we slowly (250 nl/min) infused 6 µl of 2 µg/µl 6-OHDA into the right DS. For controls, we infused vehicles (ACSF) with the same rate and same volume into the right DS without observing any damage (*Figure 1B*, *Figure 1—figure supplement 1C*). The loss of TH-positive SN neurons was quantified by unbiased stereology at 21 days and 3 months post-lesion (*Figure 1A*).

We continuously characterized the motor phenotype of the model (i.e., open field locomotion, cylinder test) until the two different end points, either 21 days post-lesion (early phase) or more than 60 days post-lesion (late phase). In the first cohorts, the mice were sacrificed for TH immuno-histochemistry in midbrain and striatum. As shown in *Figure 1B*, at the early phase, the number of TH-positive (i.e., dopaminergic) SN neurons was reduced post-6-OHDA (lower left panels) compared to vehicle-control mice (upper left panels). The right panel in *Figure 1B* displays the stereological results of TH-positive SN neurons for vehicle-treated and 6-OHDA-infused male mice 21 days after their respective infusions. The data were normalized to the stereological results of respective contra-lateral SN DA neurons. In contrast to vehicle infusions, where no ipsilateral loss of SN DA neurons was detected, the number of ipsilateral surviving SN DA neurons in the early post-6-OHDA phase was on average reduced to about 40% (*Figure 1B*; vehicle, N=9: 100 ± 2.64%; 6-OHDA, N=9: 36.85 ± 3.46%; p<0.0001, Mann–Whitney test). We also carried out the unbiased stereology for TH-positive SN neurons in the late phase and detected a very similar degree of loss (*Figure 1—figure supplement 1C*, vehicle: 102.6 ± 7.9%; 6-OHDA: 39.96 ± 3.7%; p=0.0262, Mann–Whitney test). Importantly, these results indicated that SN DA cell loss was non-progressive throughout our observation period of more than 2 months. In addition, we analyzed the axonal compartment of DA midbrain neurons by determining striatal TH-optical densities, both for early and late phase (*Figure 1—figure supplement 1A and B*). Similar to the midbrain DA cell body counts, we found a stable reduction of about 50% of TH immunosignal in the ipsilateral DS, again normalized to the contralateral side (*Figure 1—figure supplement 1A* 21st day – vehicle: 98.4 ± 2.2%; 6-OHDA: 42.5 ± 3.0%; *Figure 1—figure supplement 1B* > 64 days – vehicle: 94.4 ± 2.7%; 6-OHDA: 48.3 ± 3%), both in early and late phase. By compar-ison, the ventral striatum was only mildly affected (ca. 20% reduction, *Figure 1—figure supplement 1B* 21st day – vehicle: 94.6 ± 2.0%; 6-OHDA: 81.4 ± 3.0%; *Figure 1—figure supplement 1B* > 64 days – vehicle: 97.8 ± 2.0%; 6-OHDA: 75.2 ± 5.0%). In summary and in accordance with previous studies, our model induced stable 60% loss of SN DA neurons, associated with a stable 50% reduction of DS TH-immunoreactivity throughout the entire observation period (*Bez et al., 2016*; *Cenci and Björklund, 2020*; *Schwarting and Huston, 1996*). This stability provided a suitable framework to identify time-dependent homeostatic changes in surviving SN DA neurons.

*Figure 1C* shows the data from continuous behavioral monitoring of unilateral 6-OHDA and vehicle-treated mice before and up to 68 days post-infusion. Based on previous studies (*Cenci and Björklund, 2020*; *Schwarting and Huston, 1996*), we focused on the dynamics of drug-free sponta-neous turning behavior during open field locomotion (*Figure 1C*, left panels). In contrast to vehicle-infused mice, which displayed a stable symmetric ratio (ca. 50%) of ipsi- to contralateral turning throughout the entire experiment, 6-OHDA-infused mice showed a dramatic shift to ipsilateral turning immediately after treatment (*Figure 1C*, right panel). Interestingly, ipsilateral turns occurred in long sequences of up to 40 individual turns, a pattern not observed in controls (*Figure 1—figure supple-ment 2*). Importantly, these turning sequences as well as the overall ipsilateral bias gradually recov-ered completely over a 2-month post-lesion period (two-way ANOVA, p-value across time p<0.0001, p-value across groups p<0.0001, significant difference between vehicle and 6-OHDA group till day 40, Šídák's multiple comparisons test). Analysis of turning features, such as diameter or velocity, revealed that the recovered contralateral turns were similar to turns performed by vehicle-treated mice (*Figure 1—figure supplement 3*). Also, open field locomotion, quantified as total track length per session, recovered almost completely (*Figure 1—figure supplement 1D*). However, other motor

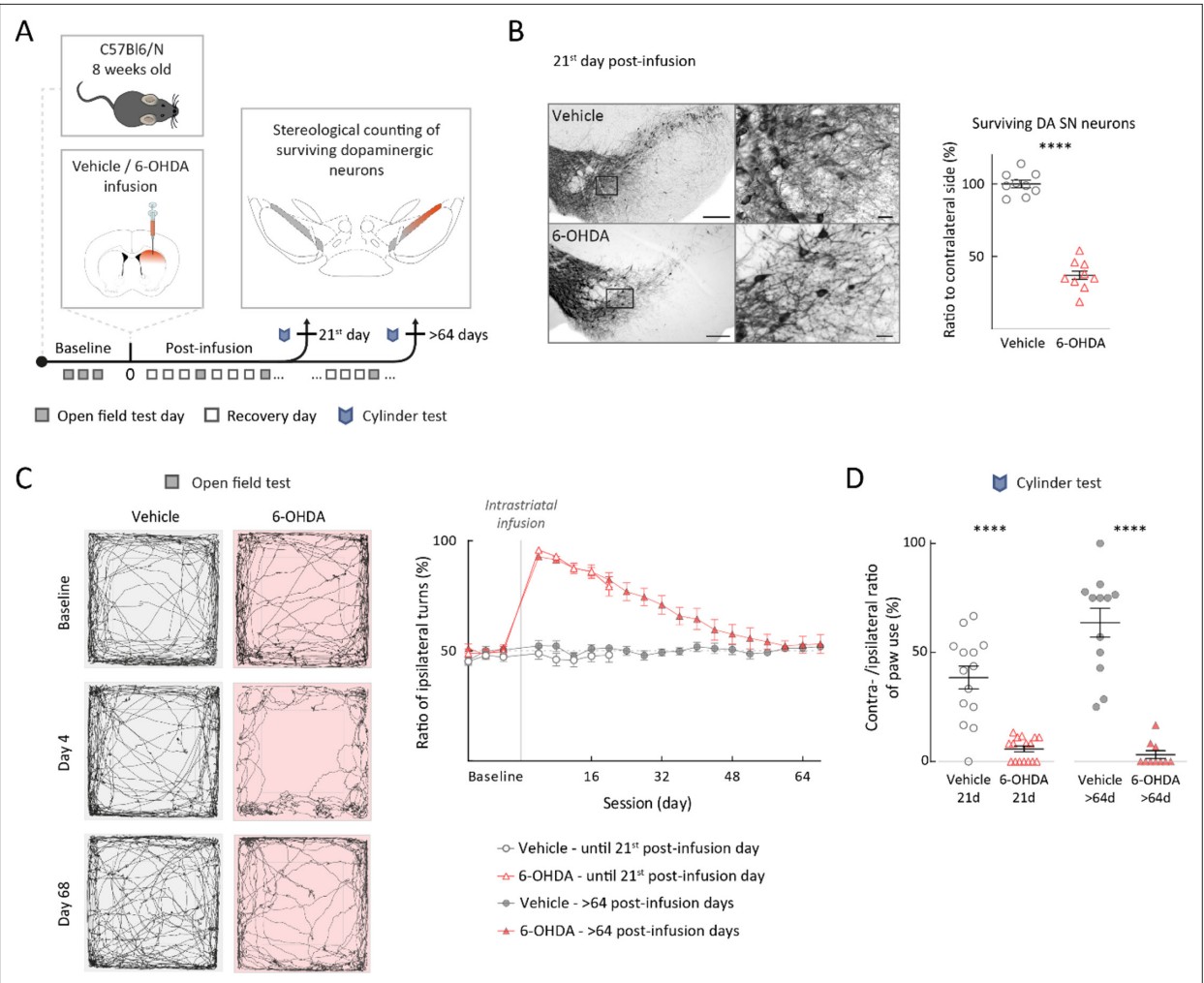

**Figure 1.** Unilateral striatal 6-hydroxy-dopamine (6-OHDA) mouse model with stable loss of substantia nigra (SN) dopamine (DA) neurons resulted in delayed partial behavioral recovery. (**A**) Experimental design, illustrating timeline of behavioral assays with termination points – groups ended either on the 21st, or later than the 64th post-infusion day. (**B**) Left panel: TH-DAB staining of the SN, ×10 magnification and ×60 magnification for vehicle and 6-OHDA-infused mouse. Scale bar left 200 µm, right 25 µm. Right panel: ratio of ipsilateral (infusion side) to contralateral side of surviving TH-positive neurons in SN at 21st post-infusion day. (**C**) Left: spontaneous locomotion of mice in open field arena for a 10 min session. Left: examples of an ACSF-infused mouse (vehicle), right example of a 6-OHDA-infused mouse at baseline (upper panels), 4th post-infusion day (middle panels), and 68th post-infusion day (lower panels). Right: ratio of ipsilateral to contralateral turning behavior for all experimental groups plotted against session days. Note recovery in the 6-OHDA-treated mice from day 4 to day 68 after initial strong turning bias (>90%). (**D**) Cylinder test quantified by ratio of contra- to ipsilateral forepaw use. Note significant loss of contralateral forepaw involvement, both at 21 and 64 days. All data are presented as mean ± standard error of mean (SEM).

The online version of this article includes the following figure supplement(s) for figure 1:

**Figure supplement 1.** Characterization of the unilateral partial striatal 6-hydroxy-dopamine (6-OHDA) model in male mice.

**Figure supplement 2.** Characterization of the unilateral partial striatal 6-hydroxy-dopamine (6-OHDA) model in female mice.

**Figure supplement 3.** Turning analysis during open field locomotion in the unilateral partial striatal 6-hydroxy-dopamine (6-OHDA) model.

**Figure supplement 4.** Ipsi- and contralateral turning features during open field locomotion in the unilateral partial striatal 6-hydroxy-dopamine (6-OHDA) model.

**Figure supplement 5.** Analysis of α-synuclein aggreates in the unilateral partial striatal 6-hydroxy-dopamine (6-OHDA) model.

behavior, the contralateral paw use during spontaneous rearing in the cylinder test, did not show significant recovery throughout the experimental period (*Figure 1D*) (21st post-infusion day: vehicle 38.5 ± 5.2%, 6-OHDA 5.7 ± 1.4%, p<0.0001, Mann–Whitney test; 68th post-infusion day: vehicle 63.7 ± 6.6%, 6-OHDA 3.2 ± 1.8%, p<0.0001, Mann–Whitney test). To test whether the 6-OHDA lesion

model at the investigated time points induced an α-synucleinopathy (i.e., α-synuclein aggregations and/or phospho-α-synuclein), we included a multilabeling immunohistochemistry analysis and utilized a viral α-synuclein injection in mice as a positive control group. Notably, no α-synuclein aggregates or phospho-α-synuclein were detected in 6-OHDA-lesioned animals at either time points (*Figure 1—figure supplement 4*). Finally, repeating the continuous behavioral monitoring and the quantification of TH immunohistochemistry in midbrain and striatum with female C57Bl6N mice, infused with either 6-OHDA or vehicle, showed very similar results and gave no indications for sex differences (*Figure 1—figure supplement 5*).

To investigate the functional properties in immunohistochemically identified surviving SN DA neurons, we performed in vivo extracellular recordings combined with juxtacellular neurobiotin (NB) labeling and in vitro whole-cell patch-clamp recordings combined with neurobiotin filling for two time points: at an early time point (21st day after lesion, early phase), where a strong turning asymmetry was present, and at a later time period (>64 days post-lesion, late phase) when turning symmetry had recovered.

## Surviving SN DA neurons in the early post-6-OHDA phase exhibit impaired firing properties both in vivo and in vitro

At the early post-6-OHDA phase, we explored the in vivo spontaneous activity of surviving identified SN DA neurons in isoflurane-anesthetized mice using single-unit extracellular recordings combined with juxtacellular neurobiotin labeling and post hoc TH immunohistochemistry (see schema in *Figure 2A*). *Figure 2B* shows the in vivo extracellular electrical activity (top panel) and identification of a representative SN DA neuron (lower right panel) from a control mouse, 3 weeks after vehicle infusion. Note the typical combination of irregular single spike firing with the occurrence of transient fast burst discharges (bursts were defined by the 80/160 ms Grace criteria; see 'Materials and methods' for details) and longer pauses. The interspike interval (ISI) histogram, which captures ISIs from the entire recording time of 10 min for this cell, displays the typical features of an in vivo bursty DA neuron, that is, a bimodal ISI distribution representing intra- and interburst intervals over a broad dynamic range between ~30 and 1000 ms (lower left panel). This recorded and labeled cell was localized in the medial SN and characterized as TH-immunopositive. In comparison, *Figure 2C* displays the representative in vivo activity of a surviving and identified SN DA neuron 3 weeks after 6-OHDA infusion. Here, while irregular single-spike pacemaking was present similar to controls, fast burst events and slow pauses were almost completely absent (top panel). Indeed, the ISI histogram of this neuron, for the entire 10 min of recording, showed a compressed dynamic firing range and was well-described by a unimodal, Gaussian-like distribution without fast intraburst and slow interburst intervals (*Figure 2C*, lower left panel). This recorded and labeled cell was also localized in the medial SN and identified as TH-immunopositive.

All in vivo recorded DA neurons, like the two representative cells shown above, from the early phase were successfully juxtacellularly labeled with NB, post hoc immunohistochemically identified and localized within the ventral midbrain (*Figure 2D*, vehicle: n=28, N=15; 6-OHDA: n=18, N=16). For best comparison, we restricted our analysis to those DA midbrain neurons localized in the medial half of the SN (mSN), where it was possible to sample a sufficient number of surviving and spontaneously active DA neurons after 6-OHDA lesion. In contrast, we only managed to detect a small number of active DA neurons in the lateral SN (lSN), the most vulnerable part of the DA midbrain (*Figure 2D*). Consequently, *Figure 2E–H* compares the group data of surviving mSN DA neurons at early phase between vehicle and 6-OHDA treatment. We detected an about 60% reduction of the mean firing rates for post-6-OHDA mSN DA neurons compared to vehicle-treated controls (*Figure 2E*; vehicle: mean frequency = 7.5 ± 0.9 Hz, 6-OHDA: mean frequency = 4.7 ± 0.6 Hz, p=0.0184, Mann–Whitney test) and no significant differences in overall firing regularity (*Figure 2F*; vehicle: CV = 53.9 ± 7.8%, 6-OHDA: CV = 36.1 ± 3.7%, p=0.1694, Mann–Whitney test). Moreover, we observed a significant 10-fold reduction in burst activity – quantified as the percentage of spikes fired in bursts (SFB) and burst rate – in mSN DA neurons in the 6-OHDA group compared to vehicle controls (*Figure 2G*, vehicle: SFB = 31.2 ± 7.7; 6-OHDA: SFB = 2.2 ± 1.1%; p=0.0032, Welch's *t*-test; *Figure 2H*, vehicle: burst rate = 0.34 ± 0.09 Hz, 6-OHDA: burst rate = 0.04 ± 0.02 Hz; p=0.0066, Welch's *t*-test). This reduction was accompanied by a pronounced shift in firing pattern, characterized by a transition from busting to irregular and pacemaker-like

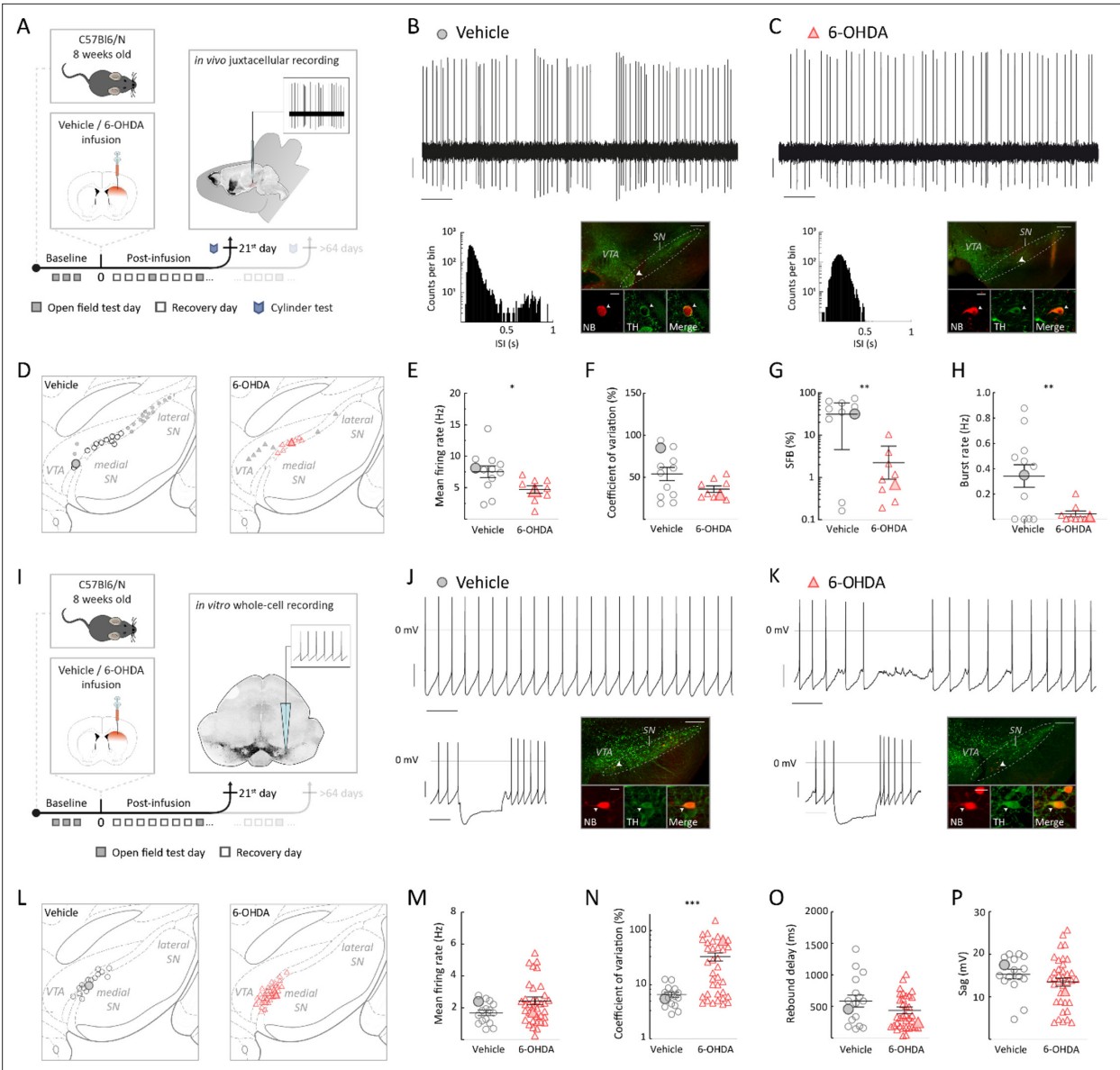

**Figure 2.** Surviving substantia nigra (SN) dopamine (DA) neurons at early post-6-hydroxy-dopamine (6-OHDA) phase exhibited a compressed dynamic range with a 10-fold decrease in in vivo bursting and a 5-fold decrease in in vitro pacemaker regularity. (**A**, **I**) Experimental design, illustrating timeline of behavioral assays, followed by terminal in vivo juxtacellular recordings (**A**) or by terminal in vitro whole-cell recordings (**I**) at the 21st post-infusion day. (**B**), (**C**) Top: 10 s original recording trace of spontaneous in vivo single-unit activity from SN DA neurons in vehicle (**B**) and 6-OHDA-infused mouse (**C**). Scale bars: 1 s, 0.2 mV. Below, left: corresponding interspike interval (ISI) histograms. Below, right: corresponding confocal images of juxtacellularly labeled and immunohistochemically identified DA neuron. Note the sparse bursting of the surviving SN DA neuron from the 6-OHDA-infused mouse at early phase. (**D**, **L**) Anatomical mapping of all extracellularly recorded and juxtacellularly labeled DA neurons (**D**), and of all in vitro recorded and filled DA neurons (**L**), projected to bregma –3.08 mm. Location of example SN DA neurons in (**B**, **C**, **J**, **K**) is highlighted. Smaller symbols represent DA neurons that have been recorded and identified, but not included in the group data analysis in (**E–H**). (**E–H**) Scatter dot-plots, showing significant decrease of in vivo mean firing rate (**E**), percentage of spikes fired in bursts (SFB) (**G**) and burst rate (**H**) and no significant differences in coefficient of variance (CV) (**F**) between the vehicle and 6-OHDA-infused mice. Note the 10-fold decrease in SFB for 6-OHDA-infused mice. (**J**, **K**) Top: 10 s original recording trace of in vitro whole-cell recording of spontaneous SN DA neuron activity in a vehicle (**J**) and 6-OHDA-infused (**K**) mouse. Scale bars: 1 s, 20 mV. Note that the 6-OHDA DA neuron has a highly irregular pacemaking. Below, left: corresponding to hyperpolarizing current injection. Below, right: confocal images of NB-filled and immunohistochemically identified DA neuron. (**M–P**) Scatter dot-plots, showing no difference in in vitro mean firing rate (**M**), rebound delay (**O**), or sag-component (**P**) and a fivefold increase in CV (**N**). Immunohistochemical imaging for all four DA neurons is displayed in ×10 and ×60 magnifications (green, TH; red, NB), scale bars: 200 μm, 20 μm. All data are presented as mean ± standard error of mean (SEM).

The online version of this article includes the following figure supplement(s) for figure 2:

**Figure supplement 1.** Analysis of in vivo electrophysiological parameters in surviving dopamine (DA) substantia nigra (SN) neuron in the early phase.

*Figure 2 continued on next page*

*Figure 2 continued*

**Figure supplement 2.** Analysis of in vitro electrophysiological parameters in surviving dopamine (DA) substantia nigra (SN) neuron in the early phase.

activity, as evidenced by the changes in the autocorrelation histogram (ACH) (***Figure 2—figure supplement 1H***).

All other analyzed firing parameters, such as intra-burst firing frequencies, identified no additional differences between the two groups (***Figure 2—figure supplement 1A–G***). In summary, we detected a rightward shift to lower firing frequencies with a corresponding compression of the dynamic in vivo firing range in mSN DA neurons surviving the 6-OHDA lesion (***Figure 2—figure supplement 1I***). Thus, in addition to DA cell loss, the dysfunction of surviving DA neurons might be a novel contributing factor to the extensive motor impairment observed during the early post-6-OHDA phase. In particular, transient bursts of DA SN neurons have been shown to provide start and stop signals for motor control (see 'Discussion' for details).

Burst discharges in DA neurons are orchestrated by the interplay of patterned synaptic inputs with their intrinsic excitability, giving rise to different types of in vivo bursting (***Otomo et al., 2020***). To identify a potential contribution of cell-autonomous changes in intrinsic excitability of surviving mSN DA neurons to their impaired in vivo dynamics, we also studied these cells in vitro in synaptic isolation (see ***Figure 2I***). ***Figure 2J*** shows a representative, stable, and regular pacemaker activity of an identified mSN DA neuron from a vehicle-infused control (lower right panel), as well as a subthreshold response to hyperpolarizing current injection (lower left panel). In contrast, pacemaker activity of mSN DA neurons post-6-OHDA infusion was unstable and characterized by intermittent periods of action potential failure (see ***Figure 2K***, upper panel). However, subthreshold responses were similar to controls (lower right panel). Analogous to the in vivo experiments, all in vitro recorded DA neurons were NB-filled, identified, and mapped, resulting in a similar medial localization of surviving SN DA neurons (***Figure 2L***). When in vitro pacemaker properties were compared between identified mSN DA neurons from the vehicle and the 6-OHDA groups (***Figure 2L***), we detected no differences in mean firing rates (***Figure 2M***, vehicle: firing rate = 1.7 ± 0.2 Hz, 6-OHDA: firing rate = 2.4 ± 0.3 Hz, p=0.1448, Mann–Whitney test), rebound delays or sag-amplitudes (***Figure 2O***, vehicle: rebound delay = 584.9 ± 95.8 ms, 6-OHDA: rebound delay = 429.6 ± 56.1 ms, p=0.1738, Mann–Whitney test; ***Figure 2P***, vehicle: sag-amplitude=15.3 ± 1.1 mV, 6-OHDA: sag-amplitude=13.5 ± 0.96 mV, p=0.1395, Mann–Whitney test). In contrast, a significant, about fivefold increase in pacemaker irregularity expressed as CV was detected (***Figure 2N***, vehicle: CV = 6.5 ± 0.7%; 6-OHDA: CV = 33.9 ± 5.8; p=0.0007, Mann–Whitney test). Detailed observation of the raw traces revealed that the increased CV was mainly mediated by a combination of intermittent firing failure, similar to the one shown in ***Figure 2K***, and episodes of elevated frequency firing. This pattern was further supported by quantitative analysis, including the ISI distribution histogram (***Figure 2—figure supplement 2H***), and the absence of significant differences between membrane voltage during active firing and voltage during pauses (***Figure 2—figure supplement 2I***). We also showed that these differences were preserved both in on-cell and whole-cell recordings, indicating the latter did not induce a differential bias in any of the two groups (e.g., different calcium buffering) (***Figure 2—figure supplement 2F and G***). Finally, other fundamental in vitro electrophysiological parameters, including afterhyperpolarization, action potential width, threshold, spike overshoot, and input resistance (***Figure 2—figure supplement 2A–E***), remained unchanged.

In short, this data demonstrates functional alterations in DA SN neurons surviving 3 weeks after a 6-OHDA infusion. These neurons exhibited a compressed dynamic range, including a 10-fold reduction of in vivo bursting. Furthermore, we observed a fivefold decrease of in vitro cell-autonomous pacemaker stability with intermittent failure of action potential firing. This compromised intrinsic in vitro excitability likely contributes to the 10-fold reduction of high-frequency in vivo firing.

In order to address the question of whether these altered properties represent stable adaptations or reflect a transient state of damage, we also studied in vivo and in vitro electrophysiology of surviving mSN DA neurons in the late phase after lesion, characterized by partial behavioral recovery of motor functions.

## Surviving mSN DA neurons in the late post-6-OHDA phase display full recovery of the in vivo dynamic range associated with an accelerated in vitro pacemaker

We recorded and analyzed surviving mSN DA neurons in the late 6-OHDA phase analogous to the early phase (compare *Figures 3A/I and 2A/I*). In contrast to the early phase, we found no significant differences in the electrophysiological in vivo properties between the post-lesional and vehicle-infused groups in the late phase (compare *Figure 3B* with *Figure 3C*). Among others (*Figure 3—figure supplement 1A–G*), this implies that mean firing frequencies, dynamic ranges, CV, SFB, and burst rates did not differ in chronically surviving mSN DA neurons compared to those in controls (*Figure 3E*, vehicle: mean firing rate = 6.9 ± 0.8 Hz; 6-OHDA: mean firing rate = 7.5 ± 1.1 Hz; p=0.6038, Mann–Whitney test; *Figure 3F*, vehicle: CV = 54.9 ± 12.2%; 6-OHDA: CV = 53.3 ± 12.6%, p>0.9999, Mann–Whitney test; *Figure 3G*, vehicle: SFB = 20.2 ± 9.1%, 6-OHDA: SFB = 22.2 ± 8.9%, p=0.8784, Mann–Whitney test; *Figure 3H*, vehicle: burst rate = 0.17 ± 0.08 Hz; 6-OHDA: burst rate = 0.30 ± 0.15 Hz, p=0.4308, Mann–Whitney test). Even though the firing pattern was nearly fully restored in the late 6-OHDA phase (*Figure 3—figure supplement 1H*), as evidenced by the reappearance of bursty firing and a close overlap in the ISI distribution histogram, a small yet significant increase in burst frequencies was still observed (*Figure 3—figure supplement 1I*). This suggests that surviving mSN DA neurons show almost full recovery of their electrophysiological properties within 2 months after the lesion.

However, our in vivo results alone cannot distinguish between two scenarios: first, a slow functional recovery process that eventually leads to return of the physiological activity range, or second, the return to the full dynamic firing range in vivo is more than simple repair to the *status quo ante* but involves a homeostatic adaptation of the functional properties of the DA neuron itself. To resolve this question, we also recorded the in vitro pacemaker properties of the surviving DA mSN neurons in the late post-6-OHDA phase. Here, we found clear evidence for allostatic adaptation of these DA neurons (i.e., the presence of new homeostatic setpoint). When comparing intrinsic pacemaker frequencies, we noticed that late 6-OHDA survivors discharged almost twofold faster compared to vehicle controls (compare *Figure 3J* with *Figure 3K*, *Figure 3M* vehicle: firing rate = 1.7 ± 0.2 Hz; 6-OHDA: firing rate = 2.7 ± 0.2 Hz, p=0.0035, Mann–Whitney test, *Figure 3—figure supplement 2H*). In contrast to speed, no significant difference in pacemaker regularity was observed (*Figure 3N*, vehicle: CV = 9.19 ± 1.30%, 6-OHDA: CV = 16.10 ± 6.66%, p=0.1245, Mann–Whitney test). Also, the sag components and the rebound delay were not significantly different (*Figure 3O*, vehicle: rebound delay = 512.8 ± 93.8 ms, 6-OHDA: rebound delay = 375.0 ± 58.9 ms, p=0.1994, Mann–Whitney test; *Figure 3P*, vehicle: sag-amplitude=16.4 ± 1.2 mV, 6-OHDA: sag-amplitude=19.1 ± 0.9 mV, p=0.0528, Mann–Whitney test). Other in vitro spike features were also not significantly different between the two groups (*Figure 3—figure supplement 2A–E*). This accelerated pacemaker phenotype was also observed in metabolically intact DA neurons recorded in the on-cell configuration but further accelerated in the whole-cell mode (*Figure 3—figure supplement 2F and G*). This selective pacemaker acceleration by conversion from the on-cell to the whole-cell configuration in the 6-OHDA group might indicate that these surviving neurons might also possess a metabolic factor that dampens the discharge rate. Although we have not carried out further mechanistic studies, altered cytosolic calcium or cyclic nucleotide concentrations might be plausible candidates.

In summary, functional recovery of the in vivo dynamic firing range of surviving mSN DA neurons was associated with homeostatic plasticity of pacemaking (i.e., an allostatic shift of pacemaker setpoint). Surviving mSN DA neurons not only recovered from instable pacemaking at the early phase but doubled their discharge rates in the late phase.

## Accelerated pacemaker caused by downregulation of Kv4.3 channels in late surviving mSN DA neurons

As Kv4.3 A-type channels have been shown to be powerful regulators of DA pacemaker frequency (*Subramaniam et al., 2014a*; *Subramaniam et al., 2014b*; *Liss et al., 2001*; *Khaliq and Bean, 2008*; *Tarfa et al., 2017*), we tested their potential role for the observed pacemaker acceleration in late phase DA mSN neurons by pharmacological occlusion experiments using 1 µM AmmTx3, a selective Kv4.3 channel blocker (see scheme in *Figure 4A*). The presence of 1 µM AmmTx3 not only accelerated discharge rates in both identified mSN DA neurons from vehicle- and 6-OHDA-infused mice (see *Figure 4BC* for representative cells; *Figure 4D* for mapping) but, importantly, also eliminated

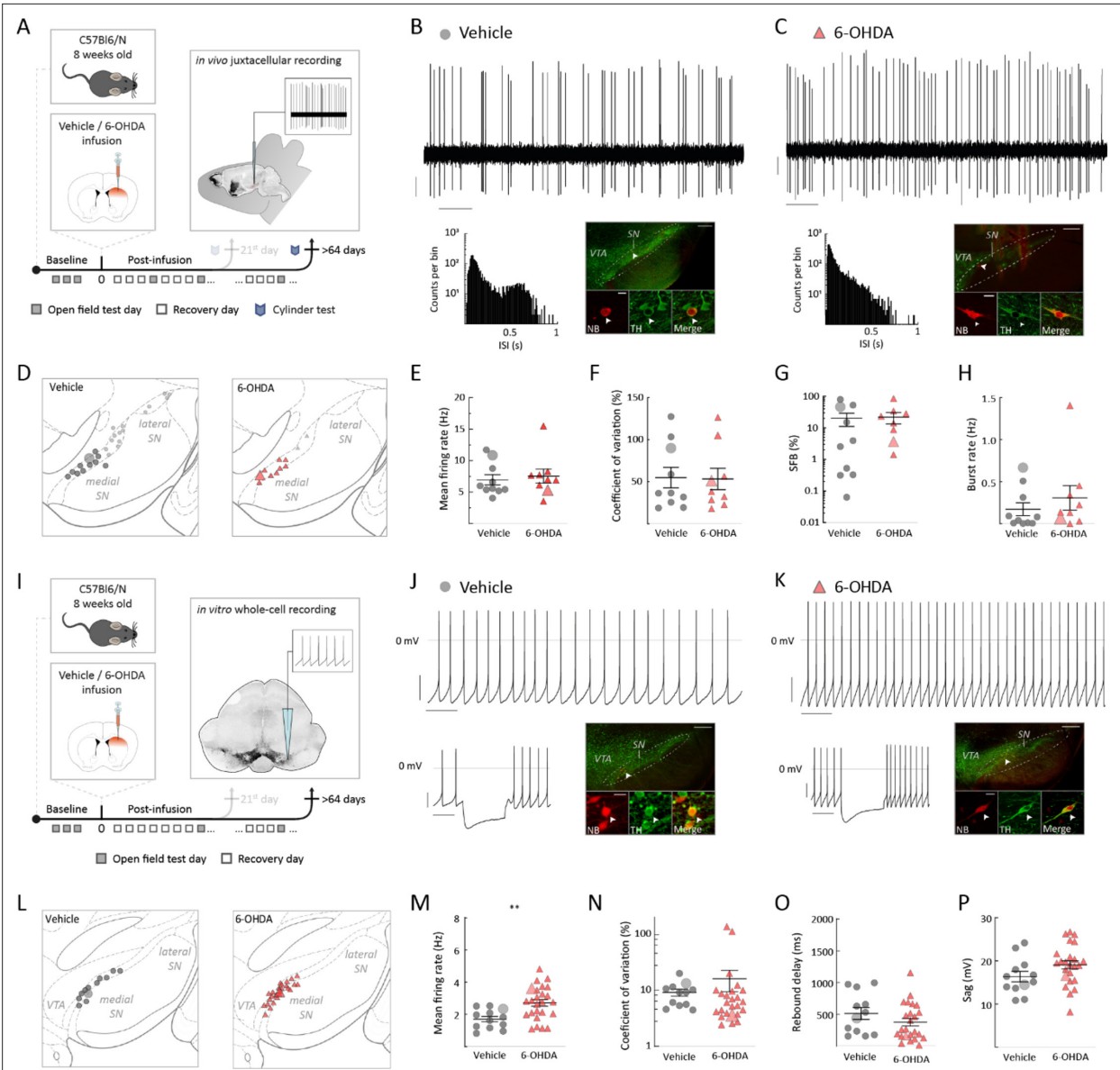

**Figure 3.** Surviving substantia nigra (SN) dopamine (DA) neurons at late 6-hydroxy-dopamine (6-OHDA) phase recovered in vivo burst firing and doubled their intrinsic pacemaker frequency in vitro. (**A**, **I**) Experimental design, illustrating timeline of behavioral assays, followed by terminal in vivo juxtacellular recordings (**A**) or by terminal in vitro whole-cell recordings (**I**) after >64 post-infusion days. (**B**, **C**) Top: 10 s original recording trace of spontaneous in vivo single-unit activity from SN DA neurons in vehicle (**B**) and 6-OHDA-infused mouse (**C**). Scale bars: 1 s, 0.2 mV. Below, left: corresponding ISI histograms. Below, right: corresponding confocal images of juxtacellularly labeled and immunohistochemically identified DA neuron. (**D**, **L**) Anatomical mapping of all extracellularly recorded and juxtacellularly labeled DA neurons (**D**), and of all in vitro recorded and filled DA neurons (**L**), projected to bregma –3.40 mm (**D**) and –3.08 mm (**L**). Location of example SN DA neurons in (**B**, **C**, **J**, **K**) is highlighted. Smaller symbols represent DA neurons that have been recorded and identified but not included in the group data analysis in (**E–H**). (**E–H**) Scatter dot-plots, showing no significant difference in mean firing rate (**E**), coefficient of variance (CV) (**F**), spikes fired in bursts (SFB) (**G**), and burst rate (**H**). (**J**, **K**) Top: 10 s original recording trace of in vitro whole-cell recording of spontaneous activity from SN DA neurons in a vehicle (**J**) and 6-OHDA-infused (**K**) mouse. Scale bars: 1 s, 20 mV. Note that the 6-OHDA DA neuron has enhanced, but regular pacemaking. Below, left, corresponding to hyperpolarizing current injection. Below, right, confocal images of NB-filled and immunohistochemically identified DA neuron. (**M–P**) Scatter dot-plots, showing doubling of the firing rate (**M**), no difference in CV (**N**), decrease in rebound delay (**O**), and increase of sag-component (**P**) for the 6-OHDA-treated mice in comparison to vehicle group. Immunohistochemical imaging for all four DA neurons is displayed in ×10 and ×60 magnifications (green, TH; red, NB), scale bars: 200 μm, 20 μm. All data are presented as mean ± standard error of mean (SEM).

The online version of this article includes the following figure supplement(s) for figure 3:

**Figure supplement 1.** Analysis of in vivo electrophysiological parameters in surviving dopamine (DA) substantia nigra (SN) neuron in the late phase.

**Figure supplement 2.** Analysis of in vitro electrophysiological parameters in surviving dopamine (DA) substantia nigra (SN) neurons in the late phase.

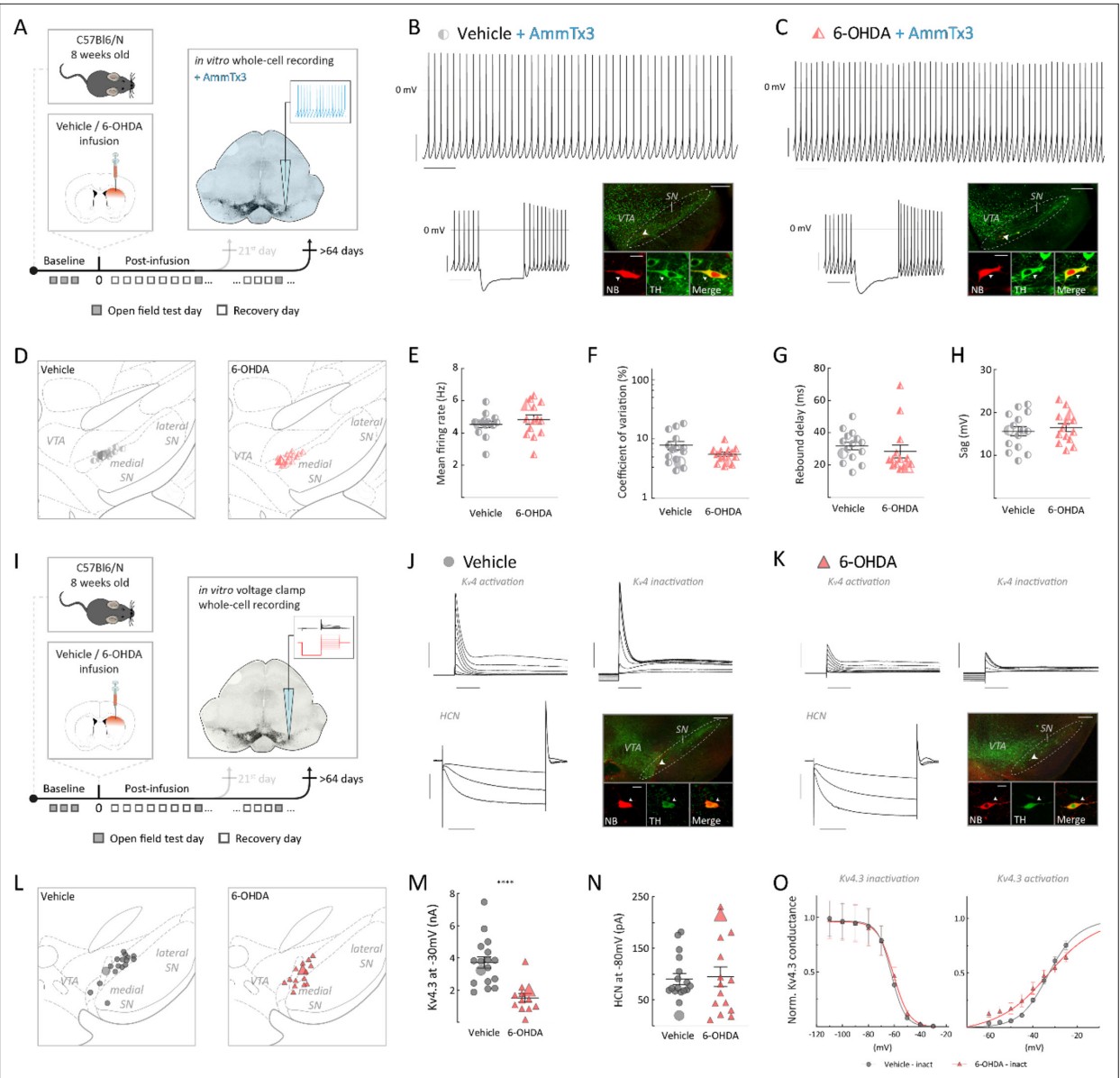

**Figure 4.** Enhanced pacemaker frequency of substantia nigra (SN) dopamine (DA) neurons at the late post-6-hydroxy-dopamine (6-OHDA) phase was mediated by Kv4.3 channel downregulation. (**A, I**) Experimental design, illustrating timeline of behavioral assays, followed by terminal in vitro whole-cell recordings under Kv4.3 channel blocker AmmTx3 (**A**) or in vitro voltage-clamp whole-cell recordings (**I**) after >64 post-infusion days. (**B, C**) Top: 10 s original recording traces of in vitro whole-cell recording of spontaneous activity from DA SN neurons in AmmTx3-preincubated slices for a vehicle (**B**) and 6-OHDA-infused (**C**) mouse. Below, left: corresponding hyperpolarizing current injection. Scale bars: 1 s, 20 mV. Below, right: confocal images of NB-filled and immunohistochemically identified DA neuron. (**D, L**) Anatomical mapping of all in vitro recorded and filled DA neurons, projected to bregma –2.92 mm (**D**) and –3.16 mm (**L**). (**E–H**) Scatter dot-plots, showing no differences in mean firing rate (**E**), coefficient of variation (**F**), rebound delay duration (**G**), and sag-amplitude (**H**). (**J, K**) Original recording trace of in vitro voltage-clamp recordings from DA SN neurons in a vehicle (**J**) and 6-OHDA-infused (**K**) mouse. Top, right: zoom-in from a Kv4.3 channel activation. Top, left: zoom-in from a Kv4.3 channel inactivation. Below, right: zoom-in from HCN-channel activation. Below, right, confocal images of NB-filled and immunohistochemically identified DA neuron. Scale bars first row: 3 nA, 100 ms. Scale bars below, left: 5 nA, 500 ms. Note the small Kv4.3 channel in/activation peak in a surviving DA neuron from a 6-OHDA mouse in comparison to the one from an ACSF-treated mouse. Immunohistochemical imaging for all four DA neurons is displayed in ×10 and ×60 magnifications (green, TH; red, NB), scale bars: 200 μm, 20 μm. All data are presented as mean ± standard error of mean (SEM). (**M**) Scatter dot-plots, showing a significant, half of maximum Kv4.3 channel conductance in surviving DA neurons. (**N**) Scatter dot-plots, showing no difference in HCN channel delta current ($I_{peak}$ – $I_{steady}$) between groups. (**O**) Normalized (Norm.) Kv4.3 conductance at different voltage steps, resulting in inactivation and activation curve for both groups. All data are presented as mean ± standard error of mean (SEM).

The online version of this article includes the following figure supplement(s) for figure 4:

*Figure 4 continued on next page*

*Figure 4 continued*

**Figure supplement 1.** Analysis of in vitro electrophysiological parameters in surviving dopamine (DA) substantia nigra (SN) neuron in the late phase during inhibition of Kv4 channels.

**Figure supplement 2.** Analysis of Kv4 whole-cell currents in surviving dopamine (DA) substantia nigra (SN) neurons in the late phase during inhibition of Kv4 channels.

**Figure supplement 3.** Analysis of HCN whole-cell currents in surviving dopamine (DA) substantia nigra (SN) neurons in the late phase.

significant rate differences between the two treatment groups (*Figure 4E*, vehicle: mean firing rate = 4.5 ± 0.2 Hz, 6-OHDA: mean firing rate = 4.8 ± 0.3 Hz, p=0.285; *Figure 4F*, vehicle: CV = 7.8 ± 1.3%, 6-OHDA: CV = 5.5 ± 0.4%, p=0.2517; both Mann–Whitney test). This result demonstrated that differences in Kv4.3 channel function were the main driver of the late post-6OHDA accelerated pacemaker phenotype. In addition, 1 μM AmmTx3 also abolished the more subtle differences in rebound delays and sag-amplitudes (*Figure 4G*, vehicle: rebound delay = 31.7 ± 2.4 ms, 6-OHDA: rebound delay = 28.4 ± 4.0 ms, p=0.0952; *Figure 4H*. vehicle: sag-amplitude=15.6 ± 1.1 mV, 6-OHDA: sag-amplitude=16.4 ± 0.9 mV, p=0.7434; both Mann–Whitney test, *Figure 4—figure supplement 1A–E and H*). Interestingly, when the experiments and analyses were repeated under pharmacological blockade of Kv4.3 channels, a significant increase in firing frequency was observed only in the vehicle group after transitioning from on-cell to whole-cell recordings. This may suggest differences in calcium buffering between the vehicle and 6-OHDA groups (*Figure 4—figure supplement 1F*). However, CV remained stable in both groups, indicating preserved firing variability (*Figure 4—figure supplement 1G*). In summary, these experiments demonstrated that reduced Kv4.3 channel function in post-6-OHDA late phase DA mSN neurons had a causal role in the phenotype of accelerated in vitro pacing.

To directly quantify the gating properties of transient outward (A-type) potassium currents mediated by Kv4.3 channels, we carried out whole-cell voltage-clamp recordings (see scheme *Figure 4I–O*). As reported before (*Liss et al., 2001*), voltage steps from negative holding potentials to the relevant subthreshold range (e.g., –60 mV to –30 mV) activated large, fast inactivating outward currents in the range of 1–10 nA in vehicle-injected controls. In contrast, the same voltage steps activated smaller fast-inactivating outward currents in post-6-OHDA late phase DA mSN neurons. These experiments demonstrated about 50% smaller maximal A-type Kv4.3 potassium currents in late surviving DA mSN neurons from 6-OHDA-treated mice compared to vehicle-treated controls (*Figure 4M*. vehicle: Kv4.3 channel current at –30 mV = 3.8 ± 0.4 nA, 6-OHDA: Kv4.3 channel current at –30 mV = 1.7 ± 0.3 nA, p<0.0001, Mann–Whitney test). When comparing voltage-dependent gating parameters, we found no changes in the biophysical channel properties apart from differences in the slope of the steady-state activation curves (*Figure 4M*, *Figure 4—figure supplement 2*).

By comparison, recordings of HCN currents activated in the voltage range between –80 mV and –120 mV showed no differences in current amplitudes and activation kinetics between the groups (*Figure 4N*, vehicle: ΔHCN@–80 mV = 90.8 ± 10.7 pA, 6-OHDA: ΔHCN@–80 mV=94.9 ± 19.1 pA, p=0.7375, Mann–Whitney test, *Figure 4—figure supplement 3*). These results substantiated the selective role of reduced Kv4.3 function in causing the accelerated pacemaker in the post-6-OHDA late phase surviving mSN DA neurons. As the reduction of Kv4.3 function might be caused by a number of different mechanisms, including reduced protein expression and/or membrane delivery of Kv4.3 subunits as well as post-translational modifications of existing Kv4.3 channel complexes (e.g., phosphorylation, redox-modification), we decided to assess Kv4.3 protein expression via immunohistochemistry.

We performed Kv4.3 immunohistochemistry and confocal imaging in vehicle and 6-OHDA-treated mice both on the ipsilateral and contralateral infusion side (see scheme in *Figure 5A*). We defined DA-selective ROIs by using TH-immunopositive areas to determine the intensity of Kv4.3-immunosignals exclusively within DA neurons (i.e., TH-signal based mask, *Figure 5—figure supplement 1A*; *Subramaniam et al., 2014a*). *Figure 5B and C* compare Kv4.3-immunoreactivities in the midbrain between the contralateral control side and the affected ipsilateral side during the late post-6-OHDA phase. Ipsilateral post-lesional TH-positive neurons showed reduced Kv4.3-immunosignals compared to the control side. Semi-quantitative comparison of TH-ROIs of the entire SN between contralateral control side and 6-OHDA-treated side revealed a significant reduction of average Kv4.3-immunosignal on the 6-OHDA-treated side of about 25% (*Figure 5D*, contralateral side n=4004, ipsilateral side, n=2645, D=0.2014, p<0.0001, two-sample Kolmogorov–Smirnov

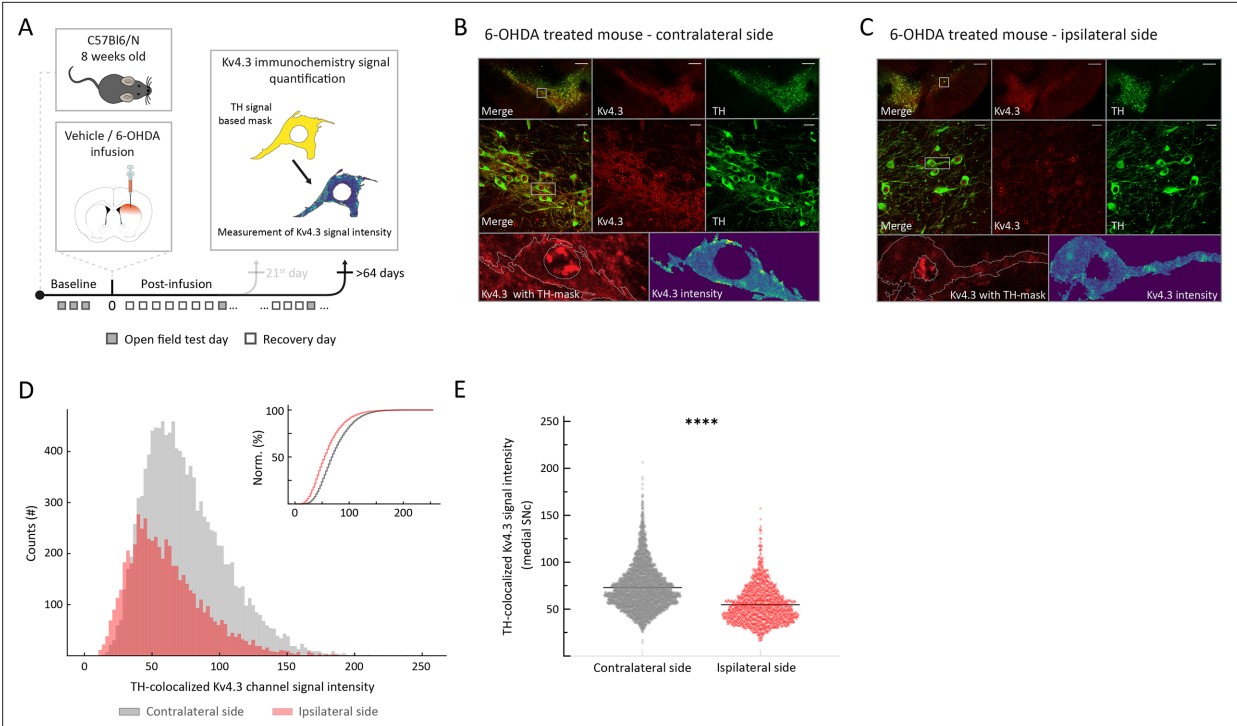

**Figure 5.** Surviving dopamine (DA) substantia nigra (SN) neurons at the late post-6-hydroxy-dopamine (6-OHDA) phase showed lower Kv4.3 channel immunohistochemical signal. (**A**) Experimental design, illustrating timeline of behavioral assays, followed by multilabeling immunohistochemistry for exploring Kv4.3-channel expression differences after >64 post-infusion days. (**B**, **C**) Top: ×4 magnification of midbrain of a 6-OHDA-infused mouse, >64 days post-lesion – contralateral side (**B**), and corresponding ipsilateral side (**C**). Middle: ×60 magnification in the highlighted area from 4× image (green, TH; red, Kv4.3). Bottom, left: zoom-in on an example ROI (highlighted in 60× image). Bottom, right: color-coded Kv4.3-channel immunohistochemical signal intensity in the example ROI. Note Kv4.3-channel signal decrease in surviving DA neurons on the ipsilateral to the injection side. Scale bars: 200 μm, 20 μm. (**D**) Histogram showing intensity of Kv4.3 immunosignals for all TH-positive ROIs, from ipsilateral, lesioned, side (in red) and from contralateral side (in gray). Inset, same data shown as a cumulative distribution. Note a clear right shift to lower intensities for the ipsilateral side. (**E**) Comparison of mean TH-colocalized Kv4.3 immunosignals from medial SN from ipsilateral, lesioned, side, and contralateral, as control side. Inset, cumulative distribution of the same data, demonstrating a rightward shift towards lower Kv4.3 immunosignal intensities (Norm.=normalized). All data are presented as mean ± standard error of mean (SEM).

The online version of this article includes the following figure supplement(s) for figure 5:

**Figure supplement 1.** Analysis of Kv4.3 immunoreactivity in surviving dopamine (DA) substantia nigra (SN) neuron in the late phase.

test). A more detailed analysis of regional differences within the SN revealed that the reduction in Kv4.3 immunoreactivity was significant for the medial SN (*Figure 5E*, contralateral side, n=4004: 73.03±0.39, ipsilateral side, n=2645: 54.8±0.49, p<0.0001, unpaired *t*-test), where we recorded Kv4-currents, but even more pronounced in the lateral parts of the SN (*Figure 5—figure supplement 1C and G*). As Kv4.3 channel subunits are expected to be mainly located on the cell membrane, we also investigated whether Kv4.3-immunosignals were differentially distributed across cell compartments. A robust, about 17% reduction of Kv4.3-immunosignals was observed in ipsilateral post-lesional TH-positive neurons both for cell membrane and cytoplasm ROIs (*Figure 5—figure supplement 1D*). In contrast, analogous quantification in vehicle-treated mice revealed no contra-ipsilateral differences in Kv4.3-immunoreactivity (*Figure 5—figure supplement 1E and F*). A comparative analysis of TH-ROI size between the contralateral and ipsilateral sides of the 6-OHDA injection revealed minimal to no differences, suggesting no significant cell shrinkage (*Figure 5—figure supplement 1B*). In summary, these immunohistochemical experiments provided evidence for a reduced expression of Kv4.3 channel proteins in late post-6OHDA mSN DA neurons. Reduced Kv4.3 protein expression is consistent with a reduction of functional Kv4.3 channels in surviving mSN DA neurons as described above and suggests that pacemaker acceleration is mediated by a reduced number of Kv4.3 channels.

## Discussion

Our study provides the first combined in vitro and in vivo electrophysiological characterization of identified DA neurons in the SN surviving a partial 6-OHDA lesion. Studying these surviving SN DA neurons at two time points, we discovered time-dependent post-6-OHDA-selective differences of firing properties both in vivo and in vitro. Early after the lesion, and coinciding with prominent behavioral impairments, we detected a selective and dramatic (about 10-fold) reduction of in vivo burst firing in identified surviving SN DA neurons. At this early time point, we also found unstable and more irregular in vitro pacemaker activity in these cells, with no differences in mean firing rates, compared to those from vehicle-infused controls.

Conversely, at a later post-lesion time point (>2 months), when partial motor recovery had occurred, we detected no differences in the in vivo firing range – including high frequency bursts – between surviving SN DA neurons post-6-OHDA and those from vehicle-infused mice. Surprisingly, the in vitro pacemaker rates in post-6-OHDA DA survivors were not only stable but almost twice as fast compared to those from vehicle controls. Finally, we pinpointed functional and protein Kv4.3 channel downregulation as the leading cause for this chronic post-lesional pacemaker acceleration, while the HCN channel function remained relatively stable. Our findings do not rule out additional changes in channel expression and function that are not directly involved (or counterbalancing each other) with regards to frequency and pattern control. Indeed, using an organotypic slice model, Wang and colleagues have identified enhanced bursting and upregulation of SK channels in DA neurons upon chronic 6-OHDA exposure (*Wang et al., 2015*). In essence, we identified a slow recovery of the dynamic in vivo firing range of surviving post-lesional DA SN neurons, linked to an allostatic acceleration of the intrinsic pacemaker by Kv4.3 channel downregulation.

Our results suggest a simple homeostatic scheme, where reducing the population size of DA SN neurons by half is at least in part compensated by a twofold increase of pacemaker speed in the remaining half of the DA SN cell population. Given the 10-fold reduction of in vivo bursting in the initial period after the lesion, it further suggests that the pacemaker setpoint rate is coupled to the in vivo high-frequency burst rate in DA mSN neurons. It is possible that a burst-related calcium signal might be detected and integrated over time to control pacemaker rate via, for example, tuning Kv4.3 expression. Recent studies have implicated the mitochondrial calcium uniporter (MCU) in coupling bursts to mitochondrial $Ca^{2+}$, acting as a calcium sensor for neuronal firing rate homeostasis (*Katsenelson et al., 2022*). Future mechanistic studies will be necessary to investigate how in vivo bursting and intrinsic pacemaking are coupled in distinct DA subpopulations. It is, however, interesting to note that the very first descriptive study of in vivo SN DA firing patterns had already found a positive correlation between the degree of in vivo burst firing and single spike firing rate (*Grace and Bunney, 1984a*; *Grace and Bunney, 1984b*). In agreement with this proposal, it was also previously found that an about fourfold reduction of in vivo burst rate via DA-selective NR1 knockout resulted in a scaled reduction of in vivo firing rate (*Zweifel et al., 2009*).

In contrast to the homeostatic role of flexible Kv4.3 expression identified in this study, we previously identified Kv4.3 channels in SN DA neurons as a pathophysiological target in a transgenic mutant (A53T-SNCA) α-synuclein mouse model (without cell loss) (*Subramaniam et al., 2014a*). In this model, we found a pacemaker acceleration caused by oxidative impairment of Kv4.3 channels. However, the activity of SN DA neurons in the α-synuclein model was accelerated also in vivo. Thus, mutant α-synuclein expression also caused a shift of the in vivo firing range of SN DA neurons toward higher frequency. While Kv4.3 subunits were downregulated at the protein level in post-lesional SN DA neurons, mutant α-synuclein-induced oxidative Kv4.3 dysfunction as well as protein upregulation (*Subramaniam et al., 2014a*). It will be interesting to test in follow-up studies if and how α-synuclein pathology affects firing and burst rate homeostasis in DA SN neurons.

Beyond pacemaker adaptations in surviving SN DA neurons, we assume that their chronic in vivo electrophysiological phenotype might also be reshaped by network level plasticity in response to DA depletion.

Our finding that the mean in vivo discharge rates were not different from controls in the presence of an accelerated intrinsic pacemaker strongly suggests a shift of the synaptic excitation-inhibition (E-I) balance toward more inhibition. Numerous studies have found altered synaptic inhibition in the DA-depleted basal ganglia (recently reviewed in *Zhang et al., 2021*). In particular, Heo and colleagues recently demonstrated a chronic E-I balance shift toward more inhibition across several PD models

including 6-OHDA (*Heo et al., 2020*). They identified a substantial contribution of additional GABA synthesis and release from reactive astrocytes in the midbrain. It has also been well-established that basal ganglia GABA neurons fire in more synchronized, hence more effective fashion after DA depletion (*Cagnan et al., 2015*; *Milosevic et al., 2018*; *Phillips et al., 2020*; *Tinkhauser et al., 2020*; *Wichmann et al., 2018*; *Evans et al., 2020*). This has recently been confirmed by elegant in vivo single-cell resolved studies using either gCAMP-based calcium monitoring or in vivo patch-clamp approaches (*Ketzef et al., 2017*; *Kravitz et al., 2010*; *Parker et al., 2018*; *Parker et al., 2016*; *Sitzia et al., 2020*). The accelerated pacemaker in response to enhanced net inhibition would also be expected within the framework of homeostatic plasticity (*Turrigiano, 2012*).

Our study has several limitations. First, we explored a single-hit, non-progressive partial lesion model, which needs to be differentiated from *state-of-the-art* PD rodent models, which develop disease-relevant α-synuclein pathology and show progressive loss of DA SN neurons (*Thakur et al., 2017*). Second, we do not differentiate between DA SN subtypes. We recently showed that lateral SN DA neurons projecting to DLS possess a distinct in vivo firing phenotype compared to medial SN DA neurons projecting to either DLS or DMS in mice (*Farassat et al., 2019*). Therefore, recording from lateral DA SN population in lesion models would be important. However, in our current partial 6-OHDA lesion model, these lateral SN DA neurons were too severely lost to allow for more than anecdotal post-lesion electrophysiological analysis (see *Figure 3*). A milder version of the current model would be needed to identify their properties and responses to partial lesion. Regarding the surviving DA neuron in the medial SN, we previously showed that they have distinct projection targets, including DLS, DMS, and lateral shell of nucleus accumbens (*Farassat et al., 2019*). In the current study, we have not aimed for identification of axonal projections to avoid potentially confounding effects of additional brain surgery. Nevertheless, this important issue should be addressed in follow-up studies by, for example, molecular subtyping approaches (*Heymann et al., 2020*; *Poulin et al., 2018*; *Saunders et al., 2018*). Third, we are aware that in vivo recordings carried out in isoflurane do not display the full dynamic spectrum of DA SN firing compared to awake, freely moving animals (we have discussed the aspect extensively in *Farassat et al., 2019*). Fourth, given the descriptive nature of our study, we do define the underlying mechanisms of the early and late changes in electrophysiological phenotypes. To what degree the early and late phenotype are caused by direct 6-OHDA toxicity, persistent damage (e.g., mitochondria), local inflammation and – we believe increasingly – by homeostatic plasticity is not addressed by this study. However, given that the in vivo phenotype recovered 2 month post-lesion, we believe it is legitimate to label the observed in vitro pacemaker adaptation homeostatic.

Finally, we would like to speculate about the possible implications of our findings for PD. In principle, cell-loss induced pacemaker plasticity, as identified here, might have a dual nature. On one hand, they might render the surviving DA SN neurons more robust in defending their phenotype and function. On the other hand, the allostatic pacemaker acceleration may amplify the innate vulnerability of an already at baseline metabolically and oxidatively challenged neuron type. The latter would be an extension of the 'stressful pacemaker' hypothesis of DA vulnerability (*Chan et al., 2007*; *Surmeier, 2007*; *Surmeier et al., 2017a*; *Surmeier et al., 2017b*) with its potential clinical implications (*Guzman et al., 2018*; *Liss and Striessnig, 2019*; *Ortner et al., 2017*; *Ortner, 2021*; *Parkinson Study Group STEADY-PD III Investigators, 2020*). In summary, we identified the homeostatic capacity and pacemaker allostasis of surviving DA SN neurons in response to cell loss.

## Materials and methods
### Animals
Male and female C57Bl/6N mice (Charles River Laboratories) were used for the study. The mice were 8 weeks old, group housed, and maintained on a 12-h light-dark cycle. All experiments and procedures involving mice were approved by the German Regierungspräsidium Darmstadt (V54-19c20/15-F40/30). In total, 152 mice were used for this study (see table below).

| | Early phase | | Late phase | |
|---|---|---|---|---|
| | Mice (#) | Cells (#) | Mice (#) | Cells (#) |

*Continued on next page*

*Continued*

| | Early phase | | Late phase | |
|---|---|---|---|---|
| In vivo juxta and behavior – vehicle | 15 | 12 – medial/ 28 – all | 12 | 10 – medial/ 26 – all |
| In vivo juxta and behavior – 6-OHDA | 16 | 9 – medial/ 16 – all | 10 | 9 – medial/ 11 – all |
| In vitro – vehicle | 3 | 16 | 3 | 12 |
| In vitro – 6-OHDA | 6 | 37 | 5 | 25 |
| In vitro and AmmTx3 – vehicle | - | - | 3 | 15 |
| In vitro and AmmTx3 – 6-OHDA | - | - | 3 | 14 |
| In vitro voltage clamp experiments – vehicle | - | - | 3 | 17 |
| In vitro voltage clamp experiments – 6-OHDA | - | | 3 | 15 |
| Female behavior experiments – vehicle, 6-OHDA | - | | 10 | |
| Female histology – vehicle, 6-OHDA | - | | 8 | |
| 6-OHDA dose response | 12 | | | |
| | Mice (#) | | Mice (#) | |
| Kv4.3 immunochemistry | 8 | | 8 | |
| SN quantification – vehicle | 9 | | 3 | |
| SN quantification – 6-OHDA | 9 | | 3 | |
| Aggregate/phospho- α-synuclein staining | 2 | | 2 | |
| Sum = 148 | | | | |

## Stereotactic 6-OHDA infusion

All surgeries were performed under general anesthesia in areflexic state. Prior to the induction of anesthesia, a premedication of 0.2 mg/kg atropine (atropine-sulfate, Braun Melsungen AG, Melsungen) was given as an intraperitoneal (i.p.) injection to stabilize circulation. Anesthesia was induced in a plastic chamber, which was flooded with 5% isoflurane (Florene, AbbVie Deutschland GmbH & Co. KG, Ludwigshafen, Germany) in pure oxygen (0.4 l/min). For maintenance of anesthesia, isoflurane was delivered through a breathing mask with a flow rate of 0.35 l/min and its concentration was regulated to 1.5–2.2% using an adjustable vaporizer (Uno, Zevenaar, Netherlands). The depth of anesthesia was controlled by testing the toe pinch reflex and the breathing rate (1–2 Hz). Body temperature (36°C) and respiration were constantly monitored. Lidocaine/prilocaine ointment (25 mg/g, Emla creme, AstraZeneca GmbH, 22876 Wedel) was applied prior to surgery and after suturing of the wound as local anesthetics. Additional analgesia was provided by subcutaneous injection of carprofen (4 mg/kg in NaCl, Rimadyl, Pfilzer GmbH, Berlin, Germany) after infusion. Eye lubricant (Visidic, Bausch and Lomb, Berlin, Germany) was used to protect eyes from desiccation.

Desipramine hydrochloride (20 mg/kg, Sigma-Aldrich) was injected i.p. 20–40 min before intracranial infusions to prevent 6-OHDA uptake by noradrenergic neurons. The desipramine solution was prepared in sterile, isotonic NaCl solution (B. Braun Melsungen AG, Germany) at the day of surgery. The infusion solutions are based on sterile artificial cerebrospinal fluid (ACSF, Harvard Apparatus, Holliston, MA, USA) with 0.02% L-ascorbic acid (used also as a vehicle solution). The 6-OHDA solution (0.2% 6-hydroxydopamine hydrochloride in ACSF with 0.02% L-ascorbic acid) was prepared at the day of infusion, stored on ice, and shielded from light.

During surgery, the animals were placed on a heating pad and were fixed in a stereotactic frame (Model 1900, Kopf Instruments, Tujunga, USA) with a stereotactic arm and a connected three-way digital positioning display. The scalp was opened by a longitudinal cut to expose the skull with bregma and lambda on display. With a centering scope (Model 1915, Kopf Instruments), the bregma-lambda distance was measured and examined for suitable anatomy (4.4 ± 0.2 mm distance). Afterward, the skull was aligned to a reference frame with a stereotaxic alignment indicator (Model 1905, Kopf Instruments) and the manipulator system was referenced to bregma.

Using a stereotaxic drill (Model 1911, Kopf Instruments) with a 500 μm diameter drill bit, a hole above the right striatum was drilled (coordinates: ML: +1.9 mm, AP: +0.5 mm to bregma). ACSF or 6-OHDA solution was loaded to a 10 μl NanoFil syringe (World Precision Instruments Inc, Sarasota, FL, USA) with a 35 G blunt needle, which was mounted on a MicroSyringe Pump (UMP3-1, World Precision Instruments) and controlled by a SYSMicro4 Controller (World Precision Instruments). Using the stereotactic arm, the needle was slowly lowered (about 750 μm/min) to a position of –2.2 mm below the brain surface (infusion site coordinates: ML: +1.9 mm, AP: +0.5 mm, DV: –2.2 mm to bregma). Anatomical references are based on *Franklin and Paxinos, 2008*. A volume of 6 μl was infused with a flow rate of 250 nl/min. Once the volume was infused, the needle rested for 5 min in that position before it was slowly moved out of the brain. Directly before and after infusion, proper functioning of the syringe system and the needle was checked. Finally, after suture, the animal was placed on a heating pad for full recovery. Oats, wet food pallets, and water were placed inside the cage to ease consumption.

## Behavioral testing
### Open field
Spontaneous locomotion (track length, wall distance, time in center, and number of rearings) and rotations of all mice were monitored in open field (50 × 50 cm, center 30 × 30 cm; red illumination, 3 lx) for 10 min in three baseline sessions and every 4th or 7th day post-infusion of ACSF/6-OHDA till the day of in vivo or in vitro experiment (e.g., 21st or >68th postoperative day). The open field was cleaned before and after each mouse with 0.1% acetic acid in distilled water. Using a video tracking system (Viewer II/III, Biobserve), spontaneous behavior was recorded and analyzed both online and offline. Data was extracted from Viewer as Excel tables and the final analysis was made with custom-made MATLAB scripts (available at https://github.com/Roeper-Lab/Kovacheva_et_al_2025).

### Cylinder test
Forelimb use during explorative activity was explored with cylinder test. The test was performed at the corresponding termination time point (20–21st and 64th post-infusion day). Mice were placed individually in a glass beaker (9 cm diameter, 19 cm height) at room light and were video recorded with a camera (Logitech HD Webcam C615) for about 5 min. No habituation was allowed before video recording. The glass cylinder was cleaned before and after every mouse with 0,1% acetic acid in distilled water. Only weight-bearing wall contacts made by each and both forelimb on the cylinder wall were scored. Wall exploration was expressed in terms of the percentage of contralateral to the infusion side (in the 6-OHDA-infused mice also impaired forepaw) to all forelimb wall contacts.

## In vivo electrophysiology
### Extracellular recording
In vivo extracellular single-unit activities of SN and VTA neurons were recorded in ACSF-infused (vehicle) and 6-OHDA-infused mice; similar procedures were used in other studies from our lab (*Farassat et al., 2019*; *Schiemann et al., 2012*; *Subramaniam et al., 2014a*). Briefly, mice were anesthetized (isoflurane; induction 4.5–5%, maintenance 1–2% in 0.4 l/min $O_2$) and placed into a stereotactic frame. The craniotomies were performed as described above to target the lateral SN (AP: –3.08 mm, ML: 1.4 mm) and medial SN (AP: –3.08 mm, ML: 0.9 mm). Borosilicate glass electrodes (10–25 MΩ; Harvard Apparatus) were made using a horizontal puller (DMZ-Universal Puller, Zeitz, Germany) and filled with 0.5 M NaCl, 10 mM HEPES (pH 7.4), and 1.5% neurobiotin (Vector Laboratories, Burlingame, CA, USA). A micromanipulator (SM-6; Luigs & Neumann, Ratingen, Germany) was used to lower the electrodes to the recording site. The single-unit activity of each neuron was recorded for at least 10 min at a sampling rate of 12.5 kHz (for firing pattern analyses), and then for another 1 min at a sampling rate of 20 kHz (for the fine analysis of AP waveforms). Signals were amplified 1000× (ELC-03M; NPI Electronics, Tamm, Germany), notch- and bandpass-filtered 0.3–5000 Hz (single-pole, 6 dB/octave, DPA-2FS, NPI Electronics) and recorded on a computer with an EPC-10 A/D converter (Heka, Lambrecht, Germany). Simultaneously, the signals were displayed on an analog oscilloscope and an audio monitor (HAMEG Instruments CombiScope HM1508; AUDIS-03/12M NPI electronics). Midbrain DA neurons were initially identified by their broad biphasic AP (>1.2 ms duration) and slow frequency (1–8 Hz)

(*Grace and Bunney, 1984b*; *Ungless and Grace, 2012*). AP duration was determined as the interval between the start of initial upward component and the minimum of following downward component.

## Juxtacellular labeling of single neurons

In order to identify the anatomical location and neurochemical identity of the recorded neurons, they were labeled post-recording with neurobiotin using the juxtacellular in vivo labeling technique (*Pinault, 1996*). Microiontophoretic currents were applied (1–10 nA positive current, 200 ms on/off pulse, ELC-03M, NPI Electronics) via the recording electrode in parallel to the monitoring of single-unit activity. Labeling was considered successful when the firing pattern of the neuron was modulated during current injection (i.e., increased activity during on-pulse and absence of activity in the off-pulse), the process was stable for at least 20 s, and was followed by the recovery of spontaneous activity. This procedure allowed for the exact mapping of the recorded DA neuron within the SN and VTA subnuclei (*Franklin and Paxinos, 2012*) using custom-written scripts in MATLAB (MathWorks, Natick, MA, USA), combined with neurochemical identification using TH-immunostaining (available at https://github.com/Roeper-Lab/Kovacheva_et_al_2025).

## In vitro electrophysiology

### Slice preparation

Animals were anesthetized by intraperitoneal injection of ketamine (250 mg/kg, Ketaset, Zoetis) and medetomidine-hydrochloride (2.5 mg/kg, Domitor, OrionPharma) prior to intracardial perfusion using ice-cold ACSF consisting of the following (in mM): 50 sucrose, 125 NaCl, 2.5 KCl, 25 NaHCO$_3$, 1.25 NaH$_2$PO$_4$, 2.5 glucose, 6 MgCl$_2$, 0.1 CaCl$_2$, and 2.96 kynurenic acid (Sigma-Aldrich), oxygenated with 95% O$_2$ and 5% CO$_2$. Rostral coronal midbrain slices (bregma: –2.92 mm to –3.16 mm) were sectioned at 250 µm using a vibrating blade microtome (VT1200s, Leica). Slices were incubated for 1 h before recordings in a 37°C bath with oxygenated extracellular solution with extra 1 µM AmmTx3, containing the following (in mM): 22.5 sucrose, 125 NaCl, 3.5 KCl, 25 NaHCO$_3$, 1.25 NaH$_2$PO$_4$, 2.5 glucose, 1.2 MgCl$_2$, and 1.2 CaCl$_2$.

### In vitro patch-clamp recordings

Slices were placed in a heated recording chamber (37°C) that was perfused with oxygenated extracellular solution at 2–4 ml/min. CNQX (20 µM; Biotrend), gabazine (SR95531, 4 µM; Biotrend), and DL-AP5 (10 µM; Cayman Chemical) were added to inhibit excitatory and inhibitory synaptic transmission. For voltage clamp recordings, TTX (500 nM; Tocris) was added to the extracellular solution. Neurons were visualized using infrared differential interference contrast videomicroscopy with a digital camera (VX55, Till Photonics) connected to an upright microscope (Axioskop 2, FSplus, Zeiss). Patch pipettes were pulled from borosilicate glass (GC150TF-10; Harvard Apparatus LTD) using a temperature-controlled, horizontal pipette puller (DMZ-Universal Puller, Zeitz). Patch pipettes (4–6 MΩ) were filled with a solution containing the following (in mM): 135 KGlu, 5 KCl, 10 HEPES, 0.1 EGTA, 5 MgCl$_2$, 0.075 CaCl$_2$, 5 NaATP, 1 LiGTP, 0.1% neurobiotin, adjusted to a pH of 7.35 with KOH. Recordings were performed using an EPC-10 patch-clamp amplifier (Heka Electronics) with a sampling rate of 20 kHz and a low-pass filter (Bessel, 5 kHz). For voltage clamp recordings, only experiments with uncompensated series resistance <10 MO were included in this study, and series resistance was electronically compensated 75%. Neurons were held at a holding potential of –40 mV to minimize HCN activation. To determine Kv4.3 activation kinetics, neurons were hyperpolarized to –80 mV for 500 ms followed by varying voltage steps from –60 mV to –20 mV in increments of 5 mV for 1 s. For inactivation kinetics, neurons were hyperpolarized from –120 mV to –20 mV in increments of 10 mV for 1 s followed by a fixed voltage step to –20 mV for 1 s. For analysis of HCN currents, neurons were hyperpolarized from –80 mV to –120 mV in increments of 20 mV for 1 s. For analysis, action potential thresholds (mV) were determined at $dV_m/dt$ >10 mV/ms.

## Immunohistochemistry

Following in vivo recordings, animals were transcardially perfused, as described previously (*Farassat et al., 2019*; *Schiemann et al., 2012*; *Subramaniam et al., 2014a*). Fixed brains were sectioned into 60 µm (midbrain) or 100 µm (forebrain) coronal sections using a vibrating microtome (VT1000S, Leica). In vitro slices were fixed in paraformaldehyde after finishing the experiment. Sections were rinsed in

PBS and then incubated in blocking solution (0.2 M PBS with 10% horse serum, 0.5% Triton X-100, 0.2% BSA). For staining with the polyclonal rabbit α-synuclein phospho S129 antibody, an antigen retrieval protocol was applied prior to blocking: sections were incubated for 30 min at 80°C in sodium citrate buffer. Afterward, the standard histological protocol continued with blocking.

Sections were incubated in carrier solution (room temperature [RT], overnight) with the following primary antibodies: polyclonal guinea pig anti-tyrosine hydroxylase (TH; 1:1000; Synaptic Systems), monoclonal mouse anti-TH (1:1000; Synaptic Systems), or polyclonal rabbit anti-TH (1:1000; Synaptic Systems); mouse anti-Kv4.3 (1:1000, Alomone Labs); polyclonal rabbit α-synuclein phospho S129 (1:200, Abcam), monoclonal rabbit anti-α-synuclein aggregate (MJFR-14-6-4-2) (1:1000, Abcam). In sequence, sections were again washed in PBS and incubated (RT, overnight) with the following secondary antibodies: goat anti-guinea pig 488, goat anti-rabbit 488, goat anti-mouse 488, goat anti-mouse 568, and goat anti-rabbit 568 (all 1:750; Thermo Fisher). Streptavidin AlexaFluor-568 and Streptavidin AlexaFluor-488 (both 1:1000; Invitrogen) were used for identifying neurobiotin-filled cells. Sections were then rinsed in PBS and mounted on slides with fluorescent mounting medium (Vectashield, Vector Laboratories).

## DAB immunocytochemistry

For DAB (3,3'-diaminobenzidine) staining procedures, a Vectastain ABC Staining Kit (Vector Laboratories) was used. Coronal sections of midbrain (30 μm) areas were cut and rinsed in PBS (3 × 10 min). Similar to previous immunolabeling procedures, unspecific antigen binding sites were blocked by incubation of the sections with blocking solution (60 min, RT). Subsequently, sections were incubated with primary antibody against TH (rabbit anti-TH) overnight, rinsed in PBS (3 × 10 min), and were incubated with biotinylated secondary antibodies (biotinylated anti-rabbit) for 2 h at RT. In parallel, an avidin-biotin complex (ABC) was formed by pre-incubation of avidin (1:1000) with biotinylated HRP (1:1000) in PBS for 2 h at RT. Sections were rinsed in PBS (3 × 10 min) prior to incubation with ABC solution (60 min, RT). Next, sections were rinsed in PBS (2 × 10 min) and Tris buffer (1 × 10 min). Finally, DAB oxidation was developed by application of 2% $H_2O_2$, 2% $NiCl_2$, and 4% DAB in Tris buffer using a DAB Substrate Kit (Vector Laboratories). $NiCl_2$ enhances sensitivity and intensity of DAB precipitation product. DAB oxidation was developed for 2–5 min and was stopped with Tris buffer once a specific high-contrast signal was detectable. Sections were rinsed in Tris buffer (3 × 10 min) and transferred onto gelatin-covered slides, air-dried overnight, and dehydrated in consecutive ascending alcohol concentrations (50%, 70%, 90% and 2 × 100%; 10 min each) followed by dehydration in xylol (2 × 100%; 10 min each). Finally, sections were mounted under glass coverslips with hardening mounting medium (Vectamount, Vector Laboratories).

## Unbiased stereology measurements

For quantification of total cell loss, TH-DAB-labeled SN DA neurons were counted using unbiased stereology based on optical dissection (*Gundersen, 1986*). In coronal sections (30 μm), the region of interest (ROI) was selected based on anatomical landmarks, including the medial lemniscus, which separates SN and adjacent VTA. Stereological counting provides unbiased data based on random, systematic sampling using an optical fractionator. This method involves counting of neurons with an optical dissector, a three-dimensional probe placed through a reference space (*Gundersen, 1986*). The optical dissector forms a focal plane with a thin depth of field through the selected sections. Objects in the focus of this focal plane are located within the reference section and are counted, while objects outside of the focal plane are not counted. On top of the optical dissector, a counting frame is applied. Counting frames ensure that all neurons have equal probabilities of being selected, regardless of shape, size, orientation, and distribution. To avoid counting capped neurons at the border of a section, an additional guard zone was deployed at the upper and lower borders of each section. DA neurons within the counting frame as well as those crossing the green line (acceptance line) were counted, while DA neurons crossing the red line (rejection line) were excluded. Moreover, only neurons with a detectable nucleus in focus within the optical dissector were counted. For quantification of total cell loss, StereoInvestigator software (V5, MicroBrightField, Colchester, USA) was used in combination with BX61 microscope (Olympus, Hamburg, Germany). The ROI was selected and marked using a low-magnification objective lens (2×, NA 0.25, Olympus) and 12–30 serial sections of 30 μm thickness were counted, covering the entire caudo-rostral extent of the SN. To count the

number of DA neurons in the SN pars compacta, a high-magnification oil-immersion objective lens (100×, NA 1.30, Olympus) (counting frame, 50×50 μm; sampling grid, 125×100 μm) was used. After counting was finished, the total number of neurons was calculated by the software using the formula described by *West, 1993*.

## Optical density measurements

Optical density measurements of TH-DAB-labeled striatal sections were performed using ImageJ software (http://rsbweb.nih.gov/ij/). Following TH-DAB labeling and TH-immunohistochemistry, images of five coronal sections (100 μm) covering the rostrocaudal axis of the striatum were captured using laser-scanning microscope (Nikon Eclipse90i, Nikon GmbH). Images were grayscale converted and mean gray values of desired striatal areas were encircled and measured. Unspecific mean gray values were measured in a defined cortical area (100 × 100 pixels) that displayed no specific TH signal due to the absence of DA innervation and were subtracted. The ventral edge of lateral ventricles served as an anatomical landmark to separate dorsal and ventral areas. For all animals, the measurement from the ipsilateral to the infusion side was divided by the contralateral side to calculate the relative optical density of the striatum.

## Immunohistochemical Kv4.3 channel signal quantification

A Nikon Eclipse 90i microscope was used for fluorescent signal detection, excitation wavelength of 488 nm for TH-signal and 568 nm for Kv4.3-channel signal. From each animal, four midbrain slices covering the caudal, intermedial, and rostral regions were selected and imaged for overview with ×4 magnification. Then ×60 magnification was used to acquire data from four areas within each SN (four images on the ipsilateral and four images on the contralateral to infusion side). All images were acquired using the same laser and camera settings. Images were exported from Nikon NIS-Elements Advanced Research (version 4.20.03) software as 8-bit TIFF files for quantification. Data was analyzed using custom-made Python 3.0 scripts with matplotlib, numpy, scimage, and scipy modules (available at https://github.com/Roeper-Lab/Kovacheva_et_al_2025). First, TH immunosignals were converted to a binary image via Otsu-thresholding algorithm to detect TH areas bigger than 400 pixels. Then, the resulting binary image was used as a mask for Kv4.3 channel immunosignal detection. For all ROIs, surface areas and mean Kv.4.3 channel signal intensity were measured. By applying erosion and dilatation algorithms on the ROIs, membrane and cytoplasm areas were segregated, allowing isolation of Kv4.3 channel immunosignal intensity for these cell compartments. Background Kv4.3 channel immunosignal was quantified in TH-immunosignal areas below the Otsu threshold. All data were then grouped according to medio-lateral and ipsi-/contralateral position for both vehicle and 6-OHDA group. Graphs and statistical analysis for this data were performed using Python custom-made scripts (available at https://github.com/Roeper-Lab/Kovacheva_et_al_2025).

## Statistical analysis

### Spike train analyses

Spike timestamps were extracted by thresholding above noise levels with IgorPro 6.02 (WaveMetrics, Lake Oswego, OR, USA). Firing pattern properties such as mean frequency, coefficient of variation (CV), and bursting measures were analyzed using custom scripts in MATLAB (available at: https://github.com/Roeper-Lab/Kovacheva_et_al_2025; copy archived at *Kovacheva, 2025*). In order to estimate burstiness and intra-burst properties, we used the burst detection methods described in *Grace and Bunney, 1984a*; *Grace and Bunney, 1984b*; *Ungless and Grace, 2012*. All non-burst-related ISIs (excluding all ISIs that followed the Grace and Bunney criteria, as well as all pre- and post-burst ISIs) were used to calculate the single spike firing frequency and single spike coefficient of variation.

For analysis of general firing patterns, ACHs were plotted using custom MATLAB scripts (available at https://github.com/Roeper-Lab/Kovacheva_et_al_2025). We used established criteria for classification of in vivo firing patterns based on visual inspection of autocorrelograms (*Farassat et al., 2019*; *Schiemann et al., 2012*; *Subramaniam et al., 2014a*): single spike oscillatory (≥3 equidistant peaks with decreasing amplitudes), single spike irregular (<3 peaks, increasing from zero approximating a steady state), bursty irregular (narrow peak with steep increase at short ISIs), and bursty oscillatory (narrow peak reflecting fast intraburst ISIs followed by clear trough and repetitive broader peaks).

## Statistics

Categorical data is represented as stacked bar graphs. To investigate the assumption of normal distribution, we performed the single-sample Kolmogorov–Smirnov test. The Mann–Whitney test, one-/two-way ANOVA was performed in non-parametric data to determine statistical significance. Categorical parameters, such as ACH-based firing pattern, were analyzed with the chi-squared test. Statistical significance level was set to $p < 0.05$. All data values are presented as means ± SEM. Statistical tests were performed using GraphPad Prism 9 (GraphPad Software, San Diego, CA, USA), MATLAB, and Python. The scatter plots are represented with median or mean ± SEM. The resulting p-values were compared with Bonferroni-corrected α-level or Tukey post hoc comparison. A value of $p \leq 0.05$ was considered to be statistically significant; $*p \leq 0.05$, $**p \leq 0.005$, $***p \leq 0.0005$. Graphs were plotted using GraphPad Prism software (9.0c), MATLAB, and Python.

## Acknowledgements

This study was supported by research grants to JR (DFG, CRC 1451). LK is an MD/PhD candidate at TransMed, Gutenberg University Mainz. We thank Beatrice Fischer and Jasmine Sonntag for technical assistance and Alexander Prinz for preliminary data on post-6-OHDA stereology.

## Additional information

### Funding

| Funder | Grant reference number | Author |
| --- | --- | --- |
| Deutsche Forschungsgemeinschaft | CRC1451 | Jochen Roeper |

The funders had no role in study design, data collection and interpretation, or the decision to submit the work for publication.

### Author contributions

Lora Kovacheva, Data curation, Formal analysis, Investigation, Methodology, Writing – original draft, Writing – review and editing; Josef Shin, Data curation, Formal analysis, Investigation, Methodology; Josefa Zaldivar-Diez, Johanna Mankel, Data curation, Formal analysis; Navid Farassat, Methodology; Kaue Machado Costa, Software, Formal analysis, Visualization, Methodology; Poonam Thakur, Conceptualization, Methodology; José A Obeso, Conceptualization, Writing – review and editing; Jochen Roeper, Conceptualization, Formal analysis, Supervision, Funding acquisition, Investigation, Methodology, Writing – original draft, Project administration, Writing – review and editing

### Author ORCIDs

Lora Kovacheva https://orcid.org/0000-0001-6999-1533
Navid Farassat https://orcid.org/0000-0002-4409-2176
Kaue Machado Costa https://orcid.org/0000-0002-5562-6495
Jochen Roeper https://orcid.org/0000-0003-2145-8742

### Ethics

All experiments and procedures involving mice were approved by the German Regierungspräsidium Darmstadt (V54-19c20/15-F40/30).

### Decision letter and Author response

Decision letter https://doi.org/10.7554/eLife.104037.sa1
Author response https://doi.org/10.7554/eLife.104037.sa2

## Additional files

### Supplementary files

MDAR checklist

## Data availability

All data have been uploaded to Dryad (https://doi.org/10.5061/dryad.76hdr7t6z).

The following dataset was generated:

| Author(s) | Year | Dataset title | Dataset URL | Database and Identifier |
|-----------|------|---------------|-------------|-------------------------|
| Jochen R | 2025 | Recovery of the full in vivo firing range in post-lesion surviving DA SN neurons associated with Kv4.3-mediated pacemaker plasticity | https://doi.org/10.5061/dryad.76hdr7t6z | Dryad Digital Repository, 10.5061/dryad.76hdr7t6z |

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
