## [Editor Report]

Kovacheva et al. investigated the electrophysiological adaptations of surviving substantia nigra dopaminergic neurons at 3 weeks and over 2 months after a 6-OHDA lesion in a Parkinsonism mouse model. They report an important early loss of burst firing and pacemaker stability, followed by a recovery of firing activity and a twofold increase in pacemaking at later stages. These compelling evidences are linked to the downregulation of the Kv4.3 potassium channel. The work provides significant insights into neuronal plasticity and will be of interest to basic neuroscientists and Parkinson's disease researchers.

---

## [Decision Letter]

**Decision letter after peer review:**

[Editors’ note: the authors submitted for reconsideration following the decision after peer review. What follows is the decision letter after the first round of review.]

Thank you for submitting the paper "Kv4.3 channel downregulation mediates chronic post-lesional pacemaker acceleration in surviving dopamine substantia nigra neurons" for consideration by *eLife*. Your article has been reviewed by 3 peer reviewers, and the evaluation has been overseen by a Reviewing Editor and a Senior Editor. The following individuals involved in review of your submission have agreed to reveal their identity: John T Williams (Reviewer #1).

Comments to the Authors:

We are sorry to say that, after consultation with the reviewers, we have decided that this work will not be considered further for publication by *eLife*, at least in the current form.

The reviewers had extensive discussions. Although not included in the individual review reports included below, the reviewers suggest that the paper might benefit from a focus on intrinsic plasticity instead of Parkinson's disease, because of the limitation associated with the 6-OHDA animal model.

While all the reviewers think that this work can be of great interest, there are major concerns: experimental conditions, data interpretation, and that importantly, Kv4.3 may not be the whole story. Given the scope of the extensive additional experiments needed, reviewers do not think it is possible that these concerns can be addressed within 3 months. Therefore, it is not appropriate for a revision per *eLife*'s policy. However, the reviewers do think that the major concerns can potentially be addressed with additional experiments, therefore, the authors are encouraged to consider a new submission in the future.

*Reviewer #1 (Recommendations for the authors):*

This manuscript examines the activity of dopamine neurons that survive a partial 6OHDA lesion. Behavioral experiments show early post lesion deficits some of which resolve over a period of 60 days. This functional recovery contrasts with the lack of any change in measures of TH expression in the dorsal lateral striatum or an increase in the number of dopamine neurons in the midbrain. Recordings from dopamine neurons both in vivo and in brain slices in the early and late stages after lesion show marked differences in firing rate and pattern. The change in firing rate is particularly evident at the late stage in experiments in brain slice experiments. The firing in vivo is somewhat normalized in lesioned animals but the pacemaker activity in brain slices is very much increased. With the administration of a potassium channel blocker that acts on the transient voltage dependent potassium conductance the firing rate in vehicle and 6OHDA treated neurons was the same. The conclusion is that the potassium conductance is increased in the late stages following 6OHDA lesion. The suggestion might be that this change in dopamine neuron activity may underlie to some degree the partial recovery of movement disorders induced by the lesion.

Comments

1. This is a tour de force that examines multiple aspects of the reaction to a partial lesion of dopamine neurons that innervate the dorso-lateral striatum. Examination of the late term plasticity both at the behavioral and electrophysiological levels is a significant contribution that illustrates the remarkable ability of the brain to adjust to trauma.

2. The identification of dopamine neurons in both in vivo and in vitro recordings is a major strength of this work.

3. The experiments used to identify the potassium conductance that is decreased after long term recovers are suggestive but may not be the whole story. Figure 3 J and K are very suggestive of a change in that conductance. The experiments in figure 4 however could also be the result of a ceiling effect on the firing rate.

Given the knowledge that pacemaker activity is the result of multiple conductances working in concert it seems that a strong statement (and even the title) that the homeostatic plasticity is the result of one specific potassium conductance might be a stretch without considerable more work that is not required for this manuscript. A change in the title and a softening of the discussion is all that is required in my opinion.

4. A thought comment. Although 6OHDA is used as a model for Parkinson's, it is not clear to me that the remarkable work done in this manuscript has any relevance to the disease. It is a superb demonstration of the plasticity of dopamine neurons and perhaps that plasticity plays a role in why the clinical onset of Parkinson's only happens after a near complete loss of dopamine neurons.

*Reviewer #2 (Recommendations for the authors):*

In this manuscript, Kovacheva et al. explored the electrophysiological features (homeostatic plasticity and adaptations) of surviving SN DA neurons at two distinct time points (3 weeks, > 2 months) from a 6-OHDA Parkinsonism mouse model. Using both single unit recording (in vivo) and whole-cell patch clamp recording, the authors report a selective loss of burst firing (in vivo) and pacemaker instability (in vitro) of mSN DA neurons early after lesion (3 weeks). They then show that the firing activity (in vivo) of mSN DA neurons is recovered late after lesion (> 2 months), which is accompanied by 2-fold increase of pacemaking activity (in vitro). They also suggest that this chronic electrophysiological adaptations in surviving mSN DA neurons are mediated by downregulation of Kv4.3 channel.

Overall, the authors do make a compelling case for homeostatic adaptations of DA neurons in a mouse model of Parkinsonism and focused on functional properties of surviving DA neurons at two distinct time points (3 weeks, > 2 months) after 6-OHDA lesion. Utilizing two different E-phys techniques, they successfully acquired physiological adaptations of surviving DA neurons from very challenging experiments. In addition, data analysis and interpretation are generally reliable and convincing. These findings can be of great interest to researchers studying the physiological alterations of DA neurons in animal PD models. However, considering the previous findings from the same group, the novelty of these findings (physiological changes from a different PD-Parkinsonism model, the same molecular mechanism) is weak and there are several major concerns (clinical relevance, experimental conditions, data interpretation) that undermine the strength of this manuscript.

Figure 1:

1. The 6-OHDA lesion model the authors used in this manuscript is basically non-progressive, while human Parkinsonism (or PD) is usually progressive. Thus, experimental conditions and outcomes in this model may be quite different from human PD pathology, which seriously limits the interpretation (and translation, clinical relevance) of the results from this manuscript.

2. Only male mice were used for this study, which also limits translational potential of the findings of this manuscript (there are many papers reporting distinct physiological and molecular alterations of DA neurons in several brain disorders between male and female mice).

3. The total volume of 6-OHDA injected into a single brain hemisphere (mouse) is extremely large (6 ul). Is this information correct? This can't be right. The usual volume of any solutions (drugs or viruses) injected into a mouse brain is less than 1 ul (sometimes 2 ul). And the speed of injection is also very high (250 nl / min). Considering these two unusual conditions, injection of 6-OHDA or vehicle solutions might nonspecifically affect or damage dorsal striatum (even though the optical density of TH in vehicle-treated mice looks OK).

4. Figure 1D: When comparing contra/ipsilateral ratio of paw use between vehicle groups (21d vs. > 64d), it looks like that the 2nd vehicle (> 64d) group shows enhanced usage of contralateral paw compared to the 1st vehicle group (21d). Is this statistically significant? Do you think this is some kind of learning effect? Moreover, although approximate, if we normalize the value of 6-OHDA group by each vehicle group at two time points, it seems like that there is a progressive reduction of contra/ipsilateral usage of paw over time by 6-OHDA lesion (approximate calculation, 21d, 5.73/38.5 = 14%; >64d, 3.167/63.67 = 4.97%). If this normalized reduction of paw usage by 6-OHDA lesion is statistically significant between two different times points, can we insist this effect is caused by stable loss of SN DA neurons? And this interpretation can be compatible with the authors' main claims (including behavioral recovery)?

Figure 2:

1. Figure 2J-K: The shapes of representative voltage sags seem to be different between vehicle and 6-OHDA groups. Are there any differences in Ih current between vehicle and 6-OHDA groups?

2. Figure 2J-K: Representative recording traces (AP firing) do not reflect well the mean firing rates of the two groups. Mean firing frequency of 6-OHDA group looks smaller than vehicle group.

Figure 3:

1. Figure 3B-C: In the sample recording traces, the firing rate of 6-OHDA group looks higher than that of vehicle group (It is better to replace those sample recording traces). This tendency is also reflected in ISI histogram graph. Most of ISI in spike events are below 0.5 s in 6-OHDA group, while there is a sizable amount of spike events showing ISI above 0.5 s in vehicle group. And this interpretation is quite different from the summary statistics in Figure 3E (no difference between the groups).

2. Figure 3K and P: The amplitude of voltage sag is significantly increased in 6-OHDA group. Have you checked Ih current of mSN DA neurons in voltage clamp and any change of protein expression of HCN channel?

3. Line 283: Supple. Figure 7F should be replaced by Supple. Figure 7F and G.

4. The number of mice used (vehicle group, N = 2 mice) for in vitro whole-cell patch clamp recording is too small.

Figure 4:

1. Figure 4A-H: Vehicle-treated (bath-application) groups are missing (at least Vehicle + Vehicle group) in the experiment.

2. Pharmacological experiment alone is weak to support the claims about molecular mechanism behind the physiological adaptations of SN DA neurons after 6-OHDA lesion. Do you think that overexpression of Kv4.3 can normalize the enhanced firing of DA neurons in 6-OHDA lesion group?

3. Figure 4J-K: The intensity of background fluorescence by Kv4.3 looks very different between contralateral and ipsilateral sides. This observation is more evident in the magnified images (bottom). In addition, there are more Kv4.3 fluorescence signal outside the DA neuron perimeter in contralateral side than ipsilateral side. Is this interpretation correct? This means that downregulation of Kv4.3 also occurs in other types of neurons surrounding mSN DA neurons in ipsilateral side?

4. It would be demanding experiments, but down-regulation of Kv4.3 can be further confirmed by other experiments including qPCR or western blot (after isolation of DA neurons)?

5. What is the nature of n number in Figure 4L-M? Isolated single DA neuron or single ROI from images?

6. Identified molecular mechanism (Kv4.3) underlying homeostatic adaptations of DA neurons is not surprising (considering the authors' previous findings, Subramaniam et al., 2014a) and does not further provide any new information (how Kv4.3 can be differently regulated in distinct types of PD models or what molecular signaling leads to down-regulation of Kv4.3 in 6-OHDA lesion model, etc.). This limitation weakens the impact of the findings provided in this manuscript.

Figure S1:

- Figure S1A-B: Given the optical density of TH signal in the representative low magnification images, the injection site of 6-OHDA is very dorsal even in the dorsal striatum. Thus, it is likely that mSN DA neurons survived because they were less exposed to 6-OHDA from the beginning. Is this interpretation right or wrong? If this is the case, those live mSN DA neurons might have been just less affected by 6-OHDA.

- Figure S1A-B: If we compare two sample images (6-OHDA ipsilateral side, 21 and >64 days), there is more optical density of TH in the dorsal side of dSTR in >64 days group than 21 days group. However, in the summary two samples comparisons, there is no recovery of TH fiber (dSTR TH optical density ratio) at all in >64 days group.

- Figure S1C: Representative TH-DAB staining images (>64 days post-infusion) are missing. Furthermore, even though the authors used non-parametric Mann-Whitney test, the number of N is too small to determine statistical significance.

- Figure S1D: The authors conclude that general locomotion, measured by track length, is recovered over time. But is this claim supported by statistical analysis? Please provide p-values (ANOVA and post-hoc).

Figure S5:

- Figure S6C: Calculated spike amplitude in mSN DA neurons is around 30 mV. Is this normal in DA neurons of 8-week-old mice?

- Figure S6H: Normalized probability and cumulative ISI distribution graphs from 6-OHDA group are left-shifted compared to vehicle group. In cumulative graph, this difference looks statistically significant (If we conduct K-S test), which means that although small, firing frequency might be up-regulated in 6-OHDA lesion group (early post-6-OHDA phase). However, there is no difference in mean firing rate between vehicle and 6-OHDA groups (Figure 2M). How can we reconcile with this discrepancy?(two sample comparison with averaged mean firing rates vs. cumulative graph incorporating all the single spike traces).

Discussion

- Line 396: Considering the general locomotion and contra/ipsilateral ratio of paw use (Figure 1D, Figure S1D), it is not sure whether we can claim that substantial behavioral recovery occurred in 6-OHDA lesion group.

- How Kv4.3 expression level can be adaptively regulated in 6-OHDA parkinsonism model? Can you suggest any potential molecular signaling?*Reviewer #3 (Recommendations for the authors):*

This manuscript by Kovacheva et al. examines motor behaviors and firing in medial substantia nigra (SN) dopaminergic (DA) neurons in a PD neurotoxin mouse model. In this partial lesion model, 6-OHDA was injected into the dorsal striatum which induced a non-progressive cell death among lateral SN DA neurons. Behavioral experiments performed at 21- and 68-days post injection showed deficits in motor function followed by partial recovery over time. To understand the mechanisms of the partial recovery, the authors tested firing in in vivo and in vitro slice experiments. At 68 days post-lesion, in vivo experiments compared firing in control to 6-OHDA treated animals and found no difference in pacemaking or burst firing, which confirms an early study of partial lesions in rats. The main novelty (reflected in the title) is that brain slice recordings from 6-OHDA treated mice showed faster pacemaking, demonstrating that adaptation occurs in intrinsic firing. Based on immunostaining, the authors then conclude that the increased firing observed in 6-OHDA treated animals can largely be explained by a downregulation of the Kv4.3 ion channel.

This is an excellent and rigorously performed study. The data are high-quality, the presentation of the data in figures and supplemental figures is clear and transparent. The conclusion that Kv4.3 protein is downregulated in the remaining dopamine neurons in 6-OHDA treated animals is solidly supported by immunohistochemical results. Regarding the weakness of the study, it is unclear whether the 25% downregulation of Kv4.3 protein is the dominant effect here. Past work from many labs has shown that Kv4.3 plays an important role in setting the DA cell firing rate which makes it an obvious choice and Kv4.3 dowregulation is likely involved here as well. However, many other channels contribute to firing in these cells and in my opinion, there are no experiments here that sufficiently address the role of other channels or show whether lower expression of Kv4.3 is the dominant effect. In addition, the authors' reliance on immunohistochemistry to provide evidence for essentially a functional mechanistic question may be problematic as the IHC may not linearly match onto functional properties of ion channels. Therefore, the authors would improve the presentation by being more balanced in their discussion of experimental findings, particularly regarding the central importance of Kv4.3 to adaptation. Despite these short comings, this is a well-executed study with many positives.

Comments:

-Quantification of Kv4.3 immunostaining suggests a 25% reduction in Kv4.3 fluorescence signal in 6-OHDA treated animals, but how this relates to recorded currents is unclear. In fact, past work from these same authors examining the A53T α-synuclein mouse model found an increase in IHC protein signal by 30% in the A53T mice while the actually A-type current was decreased by 40% due to redox (Subramaniam et al. 2014). Therefore, absolute protein levels do not provide quantitative information regarding the amplitude, voltage-dependence or dynamics of the conductances. Here, recordings of A-type currents would have provided basic needed information. Given the authors past work and expertise, it was surprising to see that these experiments were not performed. That said, the authors should address this concern either with new experiments or in words with additional text.

-Role of other conductances should be better considered. The authors find a ~ 30% increase in sag voltage suggesting an increase in Ih. An increase in Ih alone would speed firing and shorten rebound delays. However, the authors largely ignore this increase in Ih, instead referring to it as a 'subtle difference' which seems arbitrary. There are no experiments that directly test the contribution of Ih. Therefore, it seems unjustified to exclude adaptations involving Ih in the main study conclusions while emphasizing effects of approximately the same magnitude (~25-30%) on Kv4.3.

-In the occlusion experiments, the authors show that in presence of AmmTX that the difference in sag voltage between cells from vehicle and 6-OHDA treated mice is no longer present (compare Figure 3P to 4H). However, the explanation for this effect is unclear because to my knowledge, AmmTX has little to no direct effect on HCN channels. One possibility is that the membrane potential at which the sag voltage is initially measured is not well controlled, resulting in variable recruitment of Ih. Please address this.

- The authors mention that this preparation is suitable to study "functional adaptations in a stable pool of surviving SN DA neurons." The neurons tested here have survived the 6-OHDA neurotoxin but the authors do not address whether the alterations that occur result directly from the toxin. There are several studies in the literature that have tested the effects of 6-OHDA on the electrophysiology and underlying ion channels in live functioning DA neurons. Please discuss these publications in the context of the findings here.

[Editors’ note: further revisions were suggested prior to acceptance, as described below.]

Thank you for resubmitting your work entitled "Recovery of the full in vivo firing range in post-lesion surviving DA SN neurons associated with Kv4.3-mediated pacemaker plasticity" for further consideration by *eLife*. Your revised article has been evaluated by John Huguenard (Senior Editor) and a Reviewing Editor.

The manuscript has been improved but there are some remaining issues that need to be addressed, as outlined below:

*Reviewer #1 (Recommendations for the authors):*

In the present revised manuscript Kovacheva et al. have added experiments and analysis to work examining substantia nigra dopamine cells that remain following a unilateral partial lesion using 6-hydroxydopamine. In general, data on dopaminergic plasticity following insult remains of interest in neurodegenerative disorders and of particular value is the description of a time-course of this plasticity. The time-course is linked to a gradual improvement of some motor deficits that has been previously described. The work is technically exemplary and experimental details are reported with commendable detail. However, a few remaining concerns limit my enthusiasm for the interpretations of the current work. As I have only reviewed the revision, including responses provided to the original and separate reviewers, I am hesitant to suggest additional experiments, but added analyses may go a long way in alleviating the concerns regarding data presentation and interpretation.

1. Following an injection of 6µl of 6-OHDA into the dorsal striatum, data from recording experiments focus on the medial SNc where a larger number of dopamine neurons could be examined. However, it is unclear what the state of these neurons is, and accordingly, what they are intended to model. Are the remaining medial SNc neurons cells that took up 6-OHDA, either at levels similar to degenerated cells or at markedly reduced levels owing to their projection pattern, and were simply able to overcome the insult? Or are these neurons assumed to have not been exposed to any 6-OHDA in terminal regions and are therefore undergoing homeostatic plasticity exclusively due to the loss of cells that are more dorsal/lateral? I think it would be helpful if the text on this topic (lines 361-363) was instead moved to the introduction with a clear statement regarding the lack of longer-term plasticity at the single cell level in response to a 6-OHDA insult mentioned in response to original reviewer 3's third to last comment. (i.e. – what is the data that early adaptations are triggered by a 6-OHDA cascade but latent effects or additional aspects of that cascade are completely absent in later time points).

2. Regarding behavioral recovery the authors state that they have added in statistical details on the recovery of overall locomotion measured using track length. However I do not see these data in the revised manuscript at the listed lines (147-149). Instead, in the copy I have I see statistics on turning data and ipsilateral bias on lines 147-149, but no statistics on track length described in lines 152-154. The statistical data on track length recovery, or lack thereof seems reasonably important to make a stronger case that behavioral recovery is achieved in this model, otherwise the ipsilateral bias appears to be the singular behavior normalized in the current time window.

3. While the direct measurement of HCN currents is a meaningful addition to the work, it is unclear why the current was measured at -80 mV and not the maximal activation potential at -120mV (or all three potentials tested: -80 mV, -100mV, and -120mV). Judging by the representative traces the hyperpolarizing step to -80mV may simply not activate enough channels to reveal any difference between treatment and vehicle cells and an extremely informal analysis of the traces appears to show a marked difference in H-current amplitudes at -120mV. Either the maximal activation or entire voltage range should be analyzed and included, or the rationale for measuring this current at the apparently modest activating potential of -80mV should be provided.

4. Semi-related to above – the reduction of Kv4.3 current is dramatic and obvious. However it still seems – to me at least – to be an overstatement that blockade of Kv4.3 accelerating the pacemaking of vehicle cells to the level of 6-OHDA cells is sufficient to make the argument that it is the main driver of the observed plasticity in pacemaking. It would seem equally likely that Kv4.3 is one of a number of K^+^ channels responsible for regulating the speed of DA neuron pacemaking (e.g. BK, SK, Kv2.1), and modulation of these other K conductances could also accelerate vehicle pacemaking to a level similar to that seen after 6-OHDA cells. And work from a genetic model of DA cells loss did indeed show a downregulation of a number of K^+^ channel genes (González-Rodríguez et al. 2021 extended data Figure 6e). I appreciate that this is a tremendously broad concern to address experimentally, but even the examination of a single prominent K^+^ current in addition to Kv4.3 showing a selective downregulation would go a long way to strengthen the concept of 4.3 expression as a critical plasticity locus. Alternatively, the concept of Kv4.3 as one of a number of possible loci for plasticity could be added to acknowledge the possibility of multiple conductances being altered in 64d 6-OHDA cells.

*Reviewer #2 (Recommendations for the authors):*

The authors present compelling evidence for homeostatic adaptations in DA neurons in a Parkinsonism mouse model, focusing on the functional properties of surviving DA neurons at early (3 weeks) and late (>2 months) stages following a 6-OHDA lesion. By utilizing two complementary electrophysiological techniques, they successfully characterized the physiological adaptations of surviving DA neurons in experiments that are technically challenging. Moreover, their data analysis and interpretation are robust and convincing, making these findings highly relevant to researchers studying the physiological alterations in DA neurons in animal models of Parkinson's disease.

While the authors have adequately addressed most comments, some issues require further attention or correction:

- Line 109: Remove the extraneous parenthesis immediately following "6-OHDA."

- Line 119: Relocate the sample size information (N = 9) to the corresponding figure legend or include it in proper parentheses within the text.

- Line 126: Delete the phrase "please add N numbers."

- Line 156: The p-value for the 21st post-infusion day is reported as "p < 0.0001" in the manuscript, whereas it is represented as "p < 0.01" in the figure. This discrepancy should be corrected.

- Figure 4M: The p-value is displayed as "p < 0.0001," whereas other figures use the format "****." Ensure consistency in formatting throughout the manuscript.

- The presented sample traces do not appear to accurately reflect the analyzed data, particularly in Figures 2J, 2K, 2M, and 3B, 3C, 3E. The authors should verify and address this discrepancy.

- Supplementary Figure 1E: The data in this figure indicate that increasing the 6-OHDA dose leads to a higher number of surviving TH^+^ cells in the SNc, which contradicts the explanation provided by the authors. This graph requires correction and clarification.

- Figure 4O: Please add a y-axis title, such as "Normalized Kv4.3 Current," to improve clarity.

- Line 280-282: The authors did not provide an explanation for the observed further acceleration of the pacemaker phenotype in whole-cell patch clamp mode. This point should be addressed.

- The manuscript employs inconsistent figure reference formats (e.g., "Figure 2D," "Figure 2 E-H," "Figure 2E"). Please adopt a uniform reference format throughout the manuscript.

- Several supplementary figure subpanels are not mentioned in the manuscript (e.g., Supplementary Figures 5H, 6I, 7H, 10D-F). Notably, some subpanels exhibit significant differences (e.g., Supplementary Figures 9F left, 9H, 10D), which should be discussed in the manuscript.

- Figure 5D: The y-axis label is missing and should be included.

---

## [Author Response]

[Editors’ note: the authors resubmitted a revised version of the paper for consideration. What follows is the authors’ response to the first round of review.]

Reviewer #1 (Recommendations for the authors):This manuscript examines the activity of dopamine neurons that survive a partial 6OHDA lesion. Behavioral experiments show early post lesion deficits some of which resolve over a period of 60 days. This functional recovery contrasts with the lack of any change in measures of TH expression in the dorsal lateral striatum or an increase in the number of dopamine neurons in the midbrain. Recordings from dopamine neurons both in vivo and in brain slices in the early and late stages after lesion show marked differences in firing rate and pattern. The change in firing rate is particularly evident at the late stage in experiments in brain slice experiments. The firing in vivo is somewhat normalized in lesioned animals but the pacemaker activity in brain slices is very much increased. With the administration of a potassium channel blocker that acts on the transient voltage dependent potassium conductance the firing rate in vehicle and 6OHDA treated neurons was the same. The conclusion is that the potassium conductance is increased in the late stages following 6OHDA lesion. The suggestion might be that this change in dopamine neuron activity may underlie to some degree the partial recovery of movement disorders induced by the lesion.Comments1. This is a tour de force that examines multiple aspects of the reaction to a partial lesion of dopamine neurons that innervate the dorso-lateral striatum. Examination of the late term plasticity both at the behavioral and electrophysiological levels is a significant contribution that illustrates the remarkable ability of the brain to adjust to trauma.2. The identification of dopamine neurons in both in vivo and in vitro recordings is a major strength of this work.

We are grateful for the generous and constructive remarks of the reviewer.

3. The experiments used to identify the potassium conductance that is decreased after long term recovers are suggestive but may not be the whole story. Figure 3 J and K are very suggestive of a change in that conductance. The experiments in figure 4 however could also be the result of a ceiling effect on the firing rate.Given the knowledge that pacemaker activity is the result of multiple conductances working in concert it seems that a strong statement (and even the title) that the homeostatic plasticity is the result of one specific potassium conductance might be a stretch without considerable more work that is not required for this manuscript. A change in the title and a softening of the discussion is all that is required in my opinion.

We have now provided direct voltage-clamp evidence that Kv4.3 whole-cell current amplitudes are downregulated without major changes in gating (see new Figure 4J, K, M, O). Also, in contrast to the A-type currents, whole-cell HCN current amplitudes were not affected (see new Figure 4J, N).

4. A thought comment. Although 6OHDA is used as a model for Parkinson's, it is not clear to me that the remarkable work done in this manuscript has any relevance to the disease. It is a superb demonstration of the plasticity of dopamine neurons and perhaps that plasticity plays a role in why the clinical onset of Parkinson's only happens after a near complete loss of dopamine neurons.

We followed the advice of the reviewer and shifted the focus to homeostatic plasticity mechanisms of DA SN neurons. We make explicitly clear that 6OHDA is not a Parkinson model.

Reviewer #2 (Recommendations for the authors):In this manuscript, Kovacheva et al. explored the electrophysiological features (homeostatic plasticity and adaptations) of surviving SN DA neurons at two distinct time points (3 weeks, > 2 months) from a 6-OHDA Parkinsonism mouse model. Using both single unit recording (in vivo) and whole-cell patch clamp recording, the authors report a selective loss of burst firing (in vivo) and pacemaker instability (in vitro) of mSN DA neurons early after lesion (3 weeks). They then show that the firing activity (in vivo) of mSN DA neurons is recovered late after lesion (> 2 months), which is accompanied by 2-fold increase of pacemaking activity (in vitro). They also suggest that this chronic electrophysiological adaptations in surviving mSN DA neurons are mediated by downregulation of Kv4.3 channel.Overall, the authors do make a compelling case for homeostatic adaptations of DA neurons in a mouse model of Parkinsonism and focused on functional properties of surviving DA neurons at two distinct time points (3 weeks, > 2 months) after 6-OHDA lesion. Utilizing two different E-phys techniques, they successfully acquired physiological adaptations of surviving DA neurons from very challenging experiments. In addition, data analysis and interpretation are generally reliable and convincing. These findings can be of great interest to researchers studying the physiological alterations of DA neurons in animal PD models. However, considering the previous findings from the same group, the novelty of these findings (physiological changes from a different PD-Parkinsonism model, the same molecular mechanism) is weak and there are several major concerns (clinical relevance, experimental conditions, data interpretation) that undermine the strength of this manuscript.

We are grateful for the detailed reading and constructive remarks of the reviewer. However, we do not agree that the novelty of our finding is weak in the sense that mutant α-synuclein overexpression (Subramamian et al. 2014) and partial lesion (this manuscript) both activate mechanisms that converge on Kv4.3 channels. Importantly, they do so with very different results, which might have missed by the reviewer: on one hand, oxidative Kv4.3 dysfunction leads to dysregulation (in vivo and in vitro hyperexcitability), while cell loss (via 6-OHDA) is eventually compensated in vivo with firing frequency and pattern back to control levels but achieved with a new pacemaker setpoint via Kv4.3 downregulation. We have made this difference clearer in the revised manuscript.

Figure 1:1. The 6-OHDA lesion model the authors used in this manuscript is basically non-progressive, while human Parkinsonism (or PD) is usually progressive. Thus, experimental conditions and outcomes in this model may be quite different from human PD pathology, which seriously limits the interpretation (and translation, clinical relevance) of the results from this manuscript.

We followed the advice of the reviewer 1 and 2 and shifted the focus from PD to post-lesional homeostatic plasticity mechanisms of DA SN neurons. In the new submission, we make explicitly clear that 6OHDA is not a Parkinson model (non-progressive and no asyn-pathology).

2. Only male mice were used for this study, which also limits translational potential of the findings of this manuscript (there are many papers reporting distinct physiological and molecular alterations of DA neurons in several brain disorders between male and female mice).

We agree with the reviewer that potential sex-differences are highly relevant. Therefore, we have applied the model to a cohort of female mice. Regarding the temporal profile of motor impairment and recovery as well as regarding the extent of the DA cell body and axonal loss, we found no sex differences. The new data are included in the revised manuscript (see New Suppl-Figure 2).

3. The total volume of 6-OHDA injected into a single brain hemisphere (mouse) is extremely large (6 ul). Is this information correct? This can't be right. The usual volume of any solutions (drugs or viruses) injected into a mouse brain is less than 1 ul (sometimes 2 ul). And the speed of injection is also very high (250 nl / min). Considering these two unusual conditions, injection of 6-OHDA or vehicle solutions might nonspecifically affect or damage dorsal striatum (even though the optical density of TH in vehicle-treated mice looks OK).

As the reviewer correctly noted, all our experiments were controlled by vehicle infusion. Given the pump-controlled delivery of intra-striatal volumes, we indeed found no evidence for non-specific functional or structural damage. Our 6-OHDA protocol (12µg in 6µl) is no way an “extreme”– intra-striatal volume. Intrastriatal volumina ranging between 2 µl and 8 µl resulting in total 6-OHDA amount between 6 – 18 µg in the published literature in mouse are in a similar ballpark (Francardo et al. 2014) Brain; Slezia et al. (2023) Scientific reports; Mendes-Pinheiro et al. (2021) Int. J. Mol. Sci.; Stayte et al. (2020) Experimental Neurology; Lundblad et al. (2004) Neurobiol of Disease. However, somewhat alarmed by the reviewers strong statement, we explored lower volumes and drugs concentrations as alternatives. The resulting dose-response curve (see new Suppl-Figure 1E) demonstrated that our initial approach was necessary to achieve the cell loss in the range of ca. 50%.

4. Figure 1D: When comparing contra/ipsilateral ratio of paw use between vehicle groups (21d vs. > 64d), it looks like that the 2nd vehicle (> 64d) group shows enhanced usage of contralateral paw compared to the 1st vehicle group (21d). Is this statistically significant? Do you think this is some kind of learning effect? Moreover, although approximate, if we normalize the value of 6-OHDA group by each vehicle group at two time points, it seems like that there is a progressive reduction of contra/ipsilateral usage of paw over time by 6-OHDA lesion (approximate calculation, 21d, 5.73/38.5 = 14%; >64d, 3.167/63.67 = 4.97%). If this normalized reduction of paw usage by 6-OHDA lesion is statistically significant between two different times points, can we insist this effect is caused by stable loss of SN DA neurons? And this interpretation can be compatible with the authors' main claims (including behavioral recovery)?

We are grateful for the referee´s detailed reading of the data. In Author response image 1, the contra-lateral touches (“only R-touches”) are given. There are no statistical differences between the 21d post-6OHDA and the >64d post-6OHDA group. Also as shown in Author response image 1 the total amount of wall-touches (L+R+B(oth)) is stable over time without significant changes.

We have emphasized that behavioral recovery is only partial (rotational bias fully recovers, paw use does not recover). We think this might be due to larger cell loss in the lateral SN DA (compared to medial SN DA, which we have studied). Given the absence of systematic recordings from the lateral SN DA, our study cannot offer mechanistic correlates for the partial behaviroral recovery. Future studies with more selective lesions titrated to the lateral SN (DLS-projecting) will be useful in this context.

**Author response image 1. sa2fig1:** 

Figure 2:1. Figure 2J-K: The shapes of representative voltage sags seem to be different between vehicle and 6-OHDA groups. Are there any differences in Ih current between vehicle and 6-OHDA groups?

To fully quantify the involvement of ionic conductances, we have carried out additional whole-cell voltage-clamp recordings of Kv4 and HCN currents. In contrast to Kv4 currents, that were significantly reduced in DA mSN neuron from chronic post-lesional animals compared to time-matched vehicle controls (new Figure 4J,K, M,O) , we found no difference in amplitude and gating of HCN whole-cell currents (new Figure 4J, N).

2. Figure 2J-K: Representative recording traces (AP firing) do not reflect well the mean firing rates of the two groups. Mean firing frequency of 6-OHDA group looks smaller than vehicle group.

The respective example traces in Figure 2J and 2K are both well within the respective ranges of their populations (see symbols in 2M and 2N) and there is no statistically significant differences between the firing rates of the two populations (vehicle: firing rate = 1.7 ± 0.2 Hz, 6-OHDA: firing rate = 2.4 ± 0.3 Hz, p = 0.1448, Mann-Whitney test).

Figure 3:1. Figure 3B-C: In the sample recording traces, the firing rate of 6-OHDA group looks higher than that of vehicle group (It is better to replace those sample recording traces). This tendency is also reflected in ISI histogram graph. Most of ISI in spike events are below 0.5 s in 6-OHDA group, while there is a sizable amount of spike events showing ISI above 0.5 s in vehicle group. And this interpretation is quite different from the summary statistics in Figure 3E (no difference between the groups).

We show clearly for the entire data set (Figure 3E-3H) that there are no statistical differences between the in vivo firing properties of surviving DA SN neurons two month after 6-OHDA oder vehicle infusions. Accordingly, in the examples ISI histogram, which reflect 10 min of recordings, the distribution looks similar (in contrast to the early phase – see Figure 2BC). The example traces in 3BC only represent 10s and should not be overinterpreted. Therefore, we believe our representation of the data set is scientifically sound.

2. Figure 3K and P: The amplitude of voltage sag is significantly increased in 6-OHDA group. Have you checked Ih current of mSN DA neurons in voltage clamp and any change of protein expression of HCN channel?

We now provide a new data set of whole-cell patch-clamp recordings where we have also quantified the HCN-currents (see new Figure 4J, N). In contrast to Kv4.3 whole cell current amplitudes, we find no significant differences between HCN amplitudes in post6OHDA and vehicle DA SN neurons.

3. Line 283: Supple. Figure 7F should be replaced by Supple. Figure 7F and G.

This has been updated and corrected (current line 280, current Suppl. Figure 8 F-G).

4. The number of mice used (vehicle group, N = 2 mice) for in vitro whole-cell patch clamp recording is too small.

As stated above, we have increased the number of in vitro whole-cell patch clamp recordings. For in vitro recordings, the new total for post-vehicle is n=29 DA SN neurons from N=6 mice, the new total for post-6-OHDA is n=40 DA SN neurons from N=8 mice (see also Animal Table, p. 19).

Figure 4:1. Figure 4A-H: Vehicle-treated (bath-application) groups are missing (at least Vehicle + Vehicle group) in the experiment.

As AmmTx3 is water soluble and a very specific Kv4.3 blocker, we believe an additional vehicle control was not necessary, given that we also recorded the neurons under control conditions (see Figure 3 JK).

2. Pharmacological experiment alone is weak to support the claims about molecular mechanism behind the physiological adaptations of SN DA neurons after 6-OHDA lesion. Do you think that overexpression of Kv4.3 can normalize the enhanced firing of DA neurons in 6-OHDA lesion group?

As stated above, we have provided a new whole-cell voltage-clamp data set directly demonstrating the reduction of functional Kv4.3 currents in surviving post-6OHDA DA SN neurons compared to post-vehicle controls. This biophysical finding would indeed imply – as suggested by the reviewer – that additional expression of Kv4.3 channels in post6-OHDA would normalize their in vitro pacemaker rate.

3. Figure 4J-K: The intensity of background fluorescence by Kv4.3 looks very different between contralateral and ipsilateral sides. This observation is more evident in the magnified images (bottom). In addition, there are more Kv4.3 fluorescence signal outside the DA neuron perimeter in contralateral side than ipsilateral side. Is this interpretation correct? This means that downregulation of Kv4.3 also occurs in other types of neurons surrounding mSN DA neurons in ipsilateral side?

Kv4.3 protein expression is mostly detectable in TH-positive DA SN neurons in the ventral midbrain. It is expressed in somata and dendrites of these neurons. The 6OHDA induced loss of DA SN neurons therefore reduced the overall Kv4.3 immunofluorence from the DA neuropil (compare lower left high-resolution panel in Figure 5B,C). We find a low Kv4.3 immunosignal outside the TH-mask (which we termed background Kv4 signal, see Figure) which also shows are small but significant reduction. Given that Kv4.3 is exclusively expressed in DA neurons in the midbrain, our interpretation is that these regions might be low TH intensity neuropil missed by the mask-approach. We do not think that there is any evidence for a major contribution of other cell populations.

4. It would be demanding experiments, but down-regulation of Kv4.3 can be further confirmed by other experiments including qPCR or western blot (after isolation of DA neurons)?

Additional cell-specific (or single-cell) mRNA expression profiling would be feasible in principle, but given the low yield of this difficult preparation, we have prioritized voltage-clamp recordings to quantify the number of functional Kv4.3 current in single DA SN neurons. We do not think that western blotting will bring additional insights compared to high resolution Kv4.3 immunohistochemistry.

5. What is the nature of n number in Figure 4L-M? Isolated single DA neuron or single ROI from images?

In the new manuscript, that relevant data are in New Figure 5 D-E: contralateral side n = 4004; ipsilateral side, n = 2645. For analysis details – see Methods “Immunohistochemical Kv4.3 channel signal quantification“.

6. Identified molecular mechanism (Kv4.3) underlying homeostatic adaptations of DA neurons is not surprising (considering the authors' previous findings, Subramaniam et al., 2014a) and does not further provide any new information (how Kv4.3 can be differently regulated in distinct types of PD models or what molecular signaling leads to down-regulation of Kv4.3 in 6-OHDA lesion model, etc.). This limitation weakens the impact of the findings provided in this manuscript.

We believe that the reviewer has misinterpreted our previous study on Kv4.3 channels in a A53T-aSYN-mutant mouse – hence the lack of surprise. A comparison of the two studies reveals a very different role of Kv4.3

A53T-aSYN-mutant: Kv4.3 reduction (Kv4.3 current) by oxidative impairment + partial homeostatic upregulation (Kv4.3 protein) resulting in vivo hyperactivity (chronic pathophysiological state)

in contrast to

Post-6OHDA-lesion: Kv4.3 reduction (Kv4.3 current AND Kv4.3 protein) resulting in normalized in vivo activity (i.e. homeostatic success)

We have clarified these important differences in the revised manuscript.

Figure S1:- Figure S1A-B: Given the optical density of TH signal in the representative low magnification images, the injection site of 6-OHDA is very dorsal even in the dorsal striatum. Thus, it is likely that mSN DA neurons survived because they were less exposed to 6-OHDA from the beginning. Is this interpretation right or wrong? If this is the case, those live mSN DA neurons might have been just less affected by 6-OHDA.

We agree with the reviewers’ assumption that the most sensitive DA neurons in the lateral SN died at a higher proportion and more of the less vulnerable medial DA SN neurons survived and could thus be studied.

- Figure S1A-B: If we compare two sample images (6-OHDA ipsilateral side, 21 and >64 days), there is more optical density of TH in the dorsal side of dSTR in >64 days group than 21 days group. However, in the summary two samples comparisons, there is no recovery of TH fiber (dSTR TH optical density ratio) at all in >64 days group.

The reviewer missed the point that all results are normalized to the non-infused contralateral side. This said, we indeed observed no recovery of ipsilateral TH optical density.

- Figure S1C: Representative TH-DAB staining images (>64 days post-infusion) are missing. Furthermore, even though the authors used non-parametric Mann-Whitney test, the number of N is too small to determine statistical significance.

We have now included TH-DAB midbrain images. We prefer to keep the significance statement, which we believe is formally correct, without overinterpreting its result. Given the large amount of previous work is it to be expected that TH^+^ cell numbers are reduced also in the chronic state. Given that we used non-biased stereology, each data point shown here resulted from hundreds of cell counts.

- Figure S1D: The authors conclude that general locomotion, measured by track length, is recovered over time. But is this claim supported by statistical analysis? Please provide p-values (ANOVA and post-hoc).

We have provided the statistical analysis (see revised manuscript, line 147-149).

Figure S5:- Figure S6C: Calculated spike amplitude in mSN DA neurons is around 30 mV. Is this normal in DA neurons of 8-week-old mice?

The data (now in Suppl-Figure 6 C) refer to the overshoot (peak value of the action potential, see e.g. Figure 2J,K). The resulting spike amplitude is the differences between threshold and overshoot (ca. 60 mV, a standard value for DA SN neurons in adult mice; see Suppl. Figure 6, Suppl. Figure 8 and Suppl. Figure 9.)

- Figure S6H: Normalized probability and cumulative ISI distribution graphs from 6-OHDA group are left-shifted compared to vehicle group. In cumulative graph, this difference looks statistically significant (If we conduct K-S test), which means that although small, firing frequency might be up-regulated in 6-OHDA lesion group (early post-6-OHDA phase). However, there is no difference in mean firing rate between vehicle and 6-OHDA groups (Figure 2M). How can we reconcile with this discrepancy?(two sample comparison with averaged mean firing rates vs. cumulative graph incorporating all the single spike traces).

As indicated in Figure S6H the ISI distributions are significantly different (p <0.0001) but the mean firing rates are not. This is explained by the larger dispersion of the ISIs in post-6OHDA group – occurring around the observed spike failures (see Figure 2K). Accordingly, there is also a small trend in the whole-cell data towards increased CV in the post-6OHDA group (see Suppl-Figure 6 G).

Discussion- Line 396: Considering the general locomotion and contra/ipsilateral ratio of paw use (Figure 1D, Figure S1D), it is not sure whether we can claim that substantial behavioral recovery occurred in 6-OHDA lesion group.

We have toned down this statement to the more neutral expression “partial recovery” (see revised manuscript, line 366).

- How Kv4.3 expression level can be adaptively regulated in 6-OHDA parkinsonism model? Can you suggest any potential molecular signaling?

In the revised manuscript, we have clearly outlined the limitation of our study, which includes defining a molecular mechanism. We however speculate about a coupling between in vivo burst rate and its detection via e.g. mitochondrial calcium signal as a control mechanisms for Kv4.3 expression.

Reviewer #3 (Recommendations for the authors):This manuscript by Kovacheva et al. examines motor behaviors and firing in medial substantia nigra (SN) dopaminergic (DA) neurons in a PD neurotoxin mouse model. In this partial lesion model, 6-OHDA was injected into the dorsal striatum which induced a non-progressive cell death among lateral SN DA neurons. Behavioral experiments performed at 21- and 68-days post injection showed deficits in motor function followed by partial recovery over time. To understand the mechanisms of the partial recovery, the authors tested firing in in vivo and in vitro slice experiments. At 68 days post-lesion, in vivo experiments compared firing in control to 6-OHDA treated animals and found no difference in pacemaking or burst firing, which confirms an early study of partial lesions in rats. The main novelty (reflected in the title) is that brain slice recordings from 6-OHDA treated mice showed faster pacemaking, demonstrating that adaptation occurs in intrinsic firing. Based on immunostaining, the authors then conclude that the increased firing observed in 6-OHDA treated animals can largely be explained by a downregulation of the Kv4.3 ion channel.This is an excellent and rigorously performed study. The data are high-quality, the presentation of the data in figures and supplemental figures is clear and transparent. The conclusion that Kv4.3 protein is downregulated in the remaining dopamine neurons in 6-OHDA treated animals is solidly supported by immunohistochemical results. Regarding the weakness of the study, it is unclear whether the 25% downregulation of Kv4.3 protein is the dominant effect here. Past work from many labs has shown that Kv4.3 plays an important role in setting the DA cell firing rate which makes it an obvious choice and Kv4.3 dowregulation is likely involved here as well. However, many other channels contribute to firing in these cells and in my opinion, there are no experiments here that sufficiently address the role of other channels or show whether lower expression of Kv4.3 is the dominant effect. In addition, the authors' reliance on immunohistochemistry to provide evidence for essentially a functional mechanistic question may be problematic as the IHC may not linearly match onto functional properties of ion channels. Therefore, the authors would improve the presentation by being more balanced in their discussion of experimental findings, particularly regarding the central importance of Kv4.3 to adaptation. Despite these short comings, this is a well-executed study with many positives.

We thank the reviewer for detailed reading and appreciation of our study.

Comments:-Quantification of Kv4.3 immunostaining suggests a 25% reduction in Kv4.3 fluorescence signal in 6-OHDA treated animals, but how this relates to recorded currents is unclear. In fact, past work from these same authors examining the A53T α-synuclein mouse model found an increase in IHC protein signal by 30% in the A53T mice while the actually A-type current was decreased by 40% due to redox (Subramaniam et al. 2014). Therefore, absolute protein levels do not provide quantitative information regarding the amplitude, voltage-dependence or dynamics of the conductances. Here, recordings of A-type currents would have provided basic needed information. Given the authors past work and expertise, it was surprising to see that these experiments were not performed. That said, the authors should address this concern either with new experiments or in words with additional text.

We agree with the reviewer and have added whole-cell voltage clamp data that directly demonstrate the significantly reduced Kv4.3 whole-cell currents in surviving post-6-OHDA DA SNs (see new Figure 4J,K,M,O).

-Role of other conductances should be better considered. The authors find a ~ 30% increase in sag voltage suggesting an increase in Ih. An increase in Ih alone would speed firing and shorten rebound delays. However, the authors largely ignore this increase in Ih, instead referring to it as a 'subtle difference' which seems arbitrary. There are no experiments that directly test the contribution of Ih. Therefore, it seems unjustified to exclude adaptations involving Ih in the main study conclusions while emphasizing effects of approximately the same magnitude (~25-30%) on Kv4.3.

We agree with the reviewer and have added whole-cell voltage clamp data that directly demonstrate that HCN whole-cell currents in surviving post-6OHDA DA SNs are not different form controls(see new Figure 4J,N).

-In the occlusion experiments, the authors show that in presence of AmmTX that the difference in sag voltage between cells from vehicle and 6-OHDA treated mice is no longer present (compare Figure 3P to 4H). However, the explanation for this effect is unclear because to my knowledge, AmmTX has little to no direct effect on HCN channels. One possibility is that the membrane potential at which the sag voltage is initially measured is not well controlled, resulting in variable recruitment of Ih. Please address this.

As stated above, the new voltage-clamp data have demonstrated a selective reduction of Kv4.3 currents with no change of HCN currents. The AmmTX effect might have resulted from a dendritic interaction (e.g. shunting) of Kv4 and HCN channels. In light of the clear voltage-clamp results, we have not further followed this result.

- The authors mention that this preparation is suitable to study "functional adaptations in a stable pool of surviving SN DA neurons." The neurons tested here have survived the 6-OHDA neurotoxin but the authors do not address whether the alterations that occur result directly from the toxin. There are several studies in the literature that have tested the effects of 6-OHDA on the electrophysiology and underlying ion channels in live functioning DA neurons. Please discuss these publications in the context of the findings here.

The onset of the adaptive effect is many weeks after the toxin exposure. We think it therefore extremely unlike that the toxin itself has any direct effects at that later stage. The toxin might however trigger a cascade of events early – e.g. a membrane hyperpolarisation via activation of K-ATP channels (Beretta et al. 2005) – that might contribute to the observed phenotype. This possibility is added to the discussion with the reference about the direct effects of 6-OHDA (see line 361-363).

[Editors’ note: what follows is the authors’ response to the second round of review.]

The manuscript has been improved but there are some remaining issues that need to be addressed, as outlined below:Reviewer #1 (Recommendations for the authors):In the present revised manuscript Kovacheva et al. have added experiments and analysis to work examining substantia nigra dopamine cells that remain following a unilateral partial lesion using 6-hydroxydopamine. In general, data on dopaminergic plasticity following insult remains of interest in neurodegenerative disorders and of particular value is the description of a time-course of this plasticity. The time-course is linked to a gradual improvement of some motor deficits that has been previously described. The work is technically exemplary and experimental details are reported with commendable detail. However, a few remaining concerns limit my enthusiasm for the interpretations of the current work. As I have only reviewed the revision, including responses provided to the original and separate reviewers, I am hesitant to suggest additional experiments, but added analyses may go a long way in alleviating the concerns regarding data presentation and interpretation.

We sincerely appreciate the reviewer’s thorough evaluation of our revised manuscript and the acknowledgment of our work’s technical rigor and relevance to neurodegenerative disorders. We are also grateful for the reviewer’s consideration in not requiring additional experiments, and we agree that further analyses could help clarify data presentation and strengthen the interpretation of our findings (see below)

1. Following an injection of 6µl of 6-OHDA into the dorsal striatum, data from recording experiments focus on the medial SNc where a larger number of dopamine neurons could be examined. However, it is unclear what the state of these neurons is, and accordingly, what they are intended to model. Are the remaining medial SNc neurons cells that took up 6-OHDA, either at levels similar to degenerated cells or at markedly reduced levels owing to their projection pattern, and were simply able to overcome the insult? Or are these neurons assumed to have not been exposed to any 6-OHDA in terminal regions and are therefore undergoing homeostatic plasticity exclusively due to the loss of cells that are more dorsal/lateral? I think it would be helpful if the text on this topic (lines 361-363) was instead moved to the introduction with a clear statement regarding the lack of longer-term plasticity at the single cell level in response to a 6-OHDA insult mentioned in response to original reviewer 3's third to last comment. (i.e. – what is the data that early adaptations are triggered by a 6-OHDA cascade but latent effects or additional aspects of that cascade are completely absent in later time points).

Our study is the first to report the in vivo and in vitro physiology of surviving DA neurons in the medial SN at two time points. We have previously shown that DA neurons in the mSN project to three main areas of the striatum – the dorsolateral, dorsomedial and the lateral shell of nucleus accumbens (Farassat et al. 2019). Given the stable pattern of axonal loss after 6-OHDA infusion, more prominent in dorsal striatum (>50% reduction TH optical density, Suppl—Figure 1A) but also significant in ventral striatum (> 20% reduction TH optical density), demonstrates that all DA neurons – including those that survived and we recorded – had been exposed to and had taken up 6-OHDA. For the medial SN this means, that the prominent dorsal projecting DA neurons (2/3 of all DA neurons in the mSN) would have taken up more 6-OHDA and remaining ventral projecting DA SN neuron less 6-OHDA. We titrated the 6-OHDA model to about 50% overall DA cell loss to enable functional recordings of a sufficiently large set of survivors. Given the descriptive nature of our study, we cannot make mechanistic claim, which events – initially 6-OHDA toxicity directly, later persistent damage (e.g. mitochondria) or local inflammation (or combinations of these events) are driving the observed biophysical phenotypes and how this is to be separated from a purely loss-induced homeostatic plasticity. These different phases are now clearer addressed in the introduction (l. 89ff)

“Further supporting this, BerreRa et. al (BerreRa et al. 2005) proposed that early electrophysiological changes in surviving DA neurons may result directly from 6-OHDA toxicity triggering a cascade of cellular events, such as hyperpolarization mediated by K-ATP channel activation.” and in the section on limitations of study (l. 464ff).

“Fourth, given the descriptive nature of our study, we do define the underlying mechanisms of the early and late changes in electrophysiological phenotypes. To what degree the early and late phenotype are caused by direct 6-OHDA toxicity, persistent damage (e.g. mitochondria), local inflammation and – we believe increasingly – by homeostatic plasticity is not addressed by this study.”

2. Regarding behavioral recovery the authors state that they have added in statistical details on the recovery of overall locomotion measured using track length. However I do not see these data in the revised manuscript at the listed lines (147-149). Instead, in the copy I have I see statistics on turning data and ipsilateral bias on lines 147-149, but no statistics on track length described in lines 152-154. The statistical data on track length recovery, or lack thereof seems reasonably important to make a stronger case that behavioral recovery is achieved in this model, otherwise the ipsilateral bias appears to be the singular behavior normalized in the current time window.

We thank the reviewer for identifying this error. The statistical detail on the recovery of overall locomotion measured using track length have now been added to the legend of Supp-Figure 1D.

“(D) Mean track length from all mice for each open field session. Note the post-infusion drop in performed track in the 6-OHDA groups, which gradually recovers. (Infusion day marked as a thin gray line.) (two-way-ANOVA, p-value across time p<0.0001, p-value across groups p=0.0001, significant difference between vehicle and 6-OHDA group 4^th^ – 52^nd^ day and on 60^th^, Šídák's multiple comparisons test).”

3. While the direct measurement of HCN currents is a meaningful addition to the work, it is unclear why the current was measured at -80 mV and not the maximal activation potential at -120mV (or all three potentials tested: -80 mV, -100mV, and -120mV). Judging by the representative traces the hyperpolarizing step to -80mV may simply not activate enough channels to reveal any difference between treatment and vehicle cells and an extremely informal analysis of the traces appears to show a marked difference in H-current amplitudes at -120mV. Either the maximal activation or entire voltage range should be analyzed and included, or the rationale for measuring this current at the apparently modest activating potential of -80mV should be provided.

We appreciate the reviewer’s suggestion regarding an extended voltage range for measuring HCN currents. We now added the analysis of HCN-current amplitudes and activation kinetic at -100 mV and -120 mV. We also find no significant differences in HCN current properties at these more hyperpolarized membrane potentials between medial SN DA neurons from vehicle and 6-OHDA infused animals (see l. 434ff and new Supp-Figure 12AB).

4. Semi-related to above – the reduction of Kv4.3 current is dramatic and obvious. However it still seems – to me at least – to be an overstatement that blockade of Kv4.3 accelerating the pacemaking of vehicle cells to the level of 6-OHDA cells is sufficient to make the argument that it is the main driver of the observed plasticity in pacemaking. It would seem equally likely that Kv4.3 is one of a number of K^+^ channels responsible for regulating the speed of DA neuron pacemaking (e.g. BK, SK, Kv2.1), and modulation of these other K conductances could also accelerate vehicle pacemaking to a level similar to that seen after 6-OHDA cells. And work from a genetic model of DA cells loss did indeed show a downregulation of a number of K^+^ channel genes (González-Rodríguez et al. 2021 extended data Figure 6e). I appreciate that this is a tremendously broad concern to address experimentally, but even the examination of a single prominent K^+^ current in addition to Kv4.3 showing a selective downregulation would go a long way to strengthen the concept of 4.3 expression as a critical plasticity locus. Alternatively, the concept of Kv4.3 as one of a number of possible loci for plasticity could be added to acknowledge the possibility of multiple conductances being altered in 64d 6-OHDA cells.

We appreciate the reviewer’s careful consideration of our Kv4.3 occlusion experiments and the broader question of potassium channel contributions to DA SN neuron pacemaking plasticity. We fully acknowledge that Kv4.3 is one of multiple K conductances that regulate the pacemaker frequency of DA SN neurons, and that additional plastic changes in other channels cannot be ruled out. However, our results strongly support Kv4.3 downregulation as a critical driver of the observed acceleration in pacemaking.

Our pharmacological occlusion experiment using the highly selective Kv4 Blocker AmmTx3 provides direct functional evidence for the key role of Kv4.3 channels. The fact that in the presence of selective Kv4.3 inhibition, in vehicle-treated DA SN neurons and surviving 6OHDA DA SN neurons display no differences in pacemaker frequency (Figure 4 BCE) has only one simple explanation – that the main driver for the observed differences under control conditions (Figure 3JKM) is indeed Kv4.3. The reviewer voices his/her scepticism on this explanation, but does not provide an alternative explanation to explain the Kv4.3 occlusion experiment. This said, we do not rule out that other channels are also altered but not contributing to frequency control or balancing each other out. We have included this option in the text (l. 397ff).

Reviewer #2 (Recommendations for the authors):The authors present compelling evidence for homeostatic adaptations in DA neurons in a Parkinsonism mouse model, focusing on the functional properties of surviving DA neurons at early (3 weeks) and late (>2 months) stages following a 6-OHDA lesion. By utilizing two complementary electrophysiological techniques, they successfully characterized the physiological adaptations of surviving DA neurons in experiments that are technically challenging. Moreover, their data analysis and interpretation are robust and convincing, making these findings highly relevant to researchers studying the physiological alterations in DA neurons in animal models of Parkinson's disease.

We are grateful for the reviewer’s constructive feedback and encouraging comments.

While the authors have adequately addressed most comments, some issues require further attention or correction:- Line 109: Remove the extraneous parenthesis immediately following "6-OHDA."

We are grateful for the reviewer’s constructive feedback and encouraging comments.

- Line 119: Relocate the sample size information (N = 9) to the corresponding figure legend or include it in proper parentheses within the text.

The sample size (N = 9) has been added to the text (l. 126).

- Line 126: Delete the phrase "please add N numbers."

The phrase has been removed

- Line 156: The p-value for the 21st post-infusion day is reported as "p < 0.0001" in the manuscript, whereas it is represented as "p < 0.01" in the figure. This discrepancy should be corrected.

The statistic in the manuscript is the correct one. The one in the figure was corrected.

- Figure 4M: The p-value is displayed as "p < 0.0001," whereas other figures use the format "****." Ensure consistency in formatting throughout the manuscript.

We now use the format **** consistently throughout all figures including Figure 4M

- The presented sample traces do not appear to accurately reflect the analyzed data, particularly in Figures 2J, 2K, 2M, and 3B, 3C, 3E. The authors should verify and address this discrepancy.

We appreciate the reviewer’s careful assessment of the presented sample traces and the opportunity to clarify our selection criteria. The examples shown in Figures 2J, 2K, 2M, and 3B, 3C, 3E were chosen with great care, based on their anatomical location within the medial SNc, as indicated in the corresponding schematics, as well as their electrophysiological parameters. Each displayed neuron is explicitly marked within the scatter plots of the grouped data, where it is highlighted with a larger symbol for reference. Our selection process aimed to provide representative examples that closely align with the population mean values for the analyzed parameters.

- Supplementary Figure 1E: The data in this figure indicate that increasing the 6-OHDA dose leads to a higher number of surviving TH^+^ cells in the SNc, which contradicts the explanation provided by the authors. This graph requires correction and clarification.

Indeed, the y-axis title did not match the data and the description in the legend. The graph shows the loss of TH^+^ cells within the SN in relation to 6-OHDA doses. The figure has been corrected.

- Figure 4O: Please add a y-axis title, such as "Normalized Kv4.3 Current," to improve clarity.

We followed the reviewer´s suggestion and labelled the y-axis in 4O “Norm. Kv4.3 conductance”.

- Line 280-282: The authors did not provide an explanation for the observed further acceleration of the pacemaker phenotype in whole-cell patch clamp mode. This point should be addressed.

In the context of our study, Kv4.3 downregulation was identified as a key driver of the accelerated pacemaker phenotype in surviving DA SN neurons. However, additional factors, including intracellular Ca²⁺ buffering and ATP-dependent channel modulation, may contribute to the further acceleration observed in whole-cell mode. The intracellular environment of DA SN neurons contains multiple regulatory mechanisms that influence their excitability, including calcium-dependent modulation of potassium channels (e.g., SK channels), ATP-sensitive conductances, and second-messenger cascades. The pipette solution, while designed to maintain physiological conditions as much as possible, may partially disrupt these intrinsic regulatory mechanisms, thereby further accelerating pacemaking beyond what is observed in vivo. To alert the reader, we have added the following statement (l. 299ff) “This selective pacemaker acceleration by conversion from the on-cell to the whole-cell configuration in the 6-OHDA group might indicate that these surviving neurons might also possess a metabolic factor that dampens the discharge rate. Although we have not carried out further mechanistic studies, altered cytosolic calcium or cyclic nucleotide concentrations might be plausible candidates”.

- The manuscript employs inconsistent figure reference formats (e.g., "Figure 2D," "Figure 2 E-H," "Figure 2E"). Please adopt a uniform reference format throughout the manuscript.

All references to Figures were changed to the “Figure XA “ format.

- Several supplementary figure subpanels are not mentioned in the manuscript (e.g., Supplementary Figures 5H, 6I, 7H, 10D-F). Notably, some subpanels exhibit significant differences (e.g., Supplementary Figures 9F left, 9H, 10D), which should be discussed in the manuscript.

We have now referenced all Figures in the manuscript.

- Figure 5D: The y-axis label is missing and should be included.

The y-axis label of 5D has been added (#counts).